# A Cognitive Process-Inspired Architecture for Subject-Agnostic Brain Visual Decoding

**Jingyu Lu**[1]**, Haonan Wang**[1]**, Qixiang Zhang**[1]**, Xiaomeng Li**[1,2*]
[1]The Hong Kong University of Science and Technology, Hong Kong SAR, China
[2]Shenzhen Loop Area Institute, Shenzhen, China

## Abstract

Subject-agnostic brain decoding, which aims to reconstruct continuous visual experiences from fMRI without subject-specific training, holds great potential for clinical applications. However, this direction remains underexplored due to challenges in cross-subject generalization and the complex nature of brain signals. In this work, we propose Visual Cortex Flow Architecture (VCFLow), a novel hierarchical decoding framework that explicitly models the ventral-dorsal architecture of the human visual system to learn multi-dimensional representations. By disentangling and leveraging features from early visual cortex, ventral, and dorsal streams, VCFLow captures diverse and complementary cognitive information essential for visual reconstruction. Furthermore, we introduce a feature-level contrastive learning strategy to enhance the extraction of subject-invariant semantic representations, thereby enhancing subject-agnostic applicability to previously unseen subjects. Unlike conventional pipelines that need more than 12 hours of per-subject data and heavy computation, VCFLow sacrifices only 7% accuracy on average yet generates each reconstructed video in 10 seconds without any retraining, offering a fast and clinically scalable solution. The code is available at https://github.com/xmed-lab/VCFLOW.

## 1 Introduction

The world, as perceived by the human brain, unfolds as a *continuous flow* of visual experiences—not static images, but dynamic videos rich in motion, context, and meaning. Capturing this fluid nature of perception, fMRI-to-video decoding plays a critical role in revealing how the brain processes complex visual information over time, encompassing fine visual details, abstract semantic understanding, and temporal coherence. Previous research (Chen et al., 2023; Gong et al., 2024; Wang et al., 2025) has primarily focused on training and evaluating models on the same specific subjects, aiming to achieve the highest reconstruction quality. However, these methods suffer from a fundamental limitation: they often overlook subject-agnostic applicability, a factor that is even more critical in clinical settings. Specifically, when applied to new patients, these models often require more than 12 hours of subject-specific training, making them impractical for downstream tasks such as large-scale screening or clinical rehabilitation. For instance, in detecting conditions like schizophrenia, hallucinations, or cognitive impairments, subject-specific models become impractical due to their reliance on extensive retraining, which is both time-consuming and expensive. As a result, developing a **subject-agnostic model** capable of directly evaluating **previously unseen subjects** holds far greater clinical and practical value. However, this area remains largely underexplored.

To achieve subject-agnostic capability, a straightforward approach is to modify the preprocessing pipeline and encoding scheme of existing subject-specific models (Gong et al., 2024; Wang et al., 2025) to map into a shared representational space, thereby enabling a subject-agnostic setting. However, when applied to unseen subjects, these subject-specific methods performed poorly, primarily due to their inability to extract universal semantic information across subjects, as demonstrated by the results of NEURONS* in Table 1. To address this, the GLFA (Li et al., 2024) method proposed a data-level functional alignment strategy that projects data from different subjects into a universal space. However, its reliance on pretraining with fMRI data from all subjects substantially reduces its

---

*Corresponding author: eexmli@ust.hk

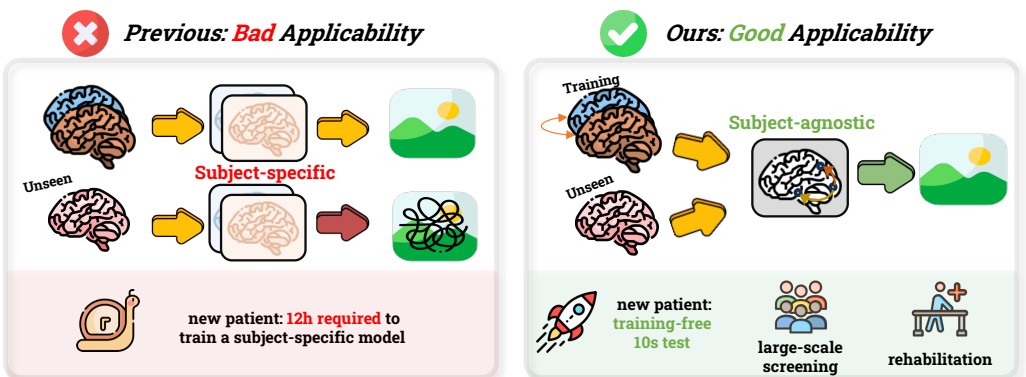

Figure 1: Former methods are typically subject-dependent, meaning that when encountering a new patient, approximately 12 hours of training are required to build a subject-specific model. Such requirements severely constrain the practical applicability and clinical utility of these approaches. By contrast, our method ensures applicability at the subject-agnostic level, allowing inference on a new patient without any additional training and requiring only about 10 seconds of testing, which provides substantial advantages for downstream tasks.

practical applicability and runs counter to the subject-agnostic paradigm. In this case, it is necessary to redesign a truly **subject-agnostic** model that enables robust decoding across new subjects at the level of cognitive features.

To this end, we introduce Visual Cortex Flow Architecture (VCFLOW), a novel subject-agnostic architecture inspired by the dual-stream mechanism of the human visual cortex (Kandel et al., 2000; Huff et al., 2018; DiCarlo et al., 2012; Dumoulin & Wandell, 2008; Yamins & DiCarlo, 2016). As illustrated in Fig. 2, human visual cognition primarily proceeds along two pathways: the *ventral stream*, extending from early vision to encode high-level semantics such as abstract concepts and object recognition, and the *dorsal stream*, extending from early vision to capture dynamic features including movement direction, velocity, and spatial transformations. Following this, we split the fMRI brain features into three components: (1) *early visual areas*, which are aligned with CLIP low-level features to capture perceptual and structural properties; (2) *ventral stream areas*, which are aligned with CLIP high-level features to learn abstract semantics; and (3) *dorsal stream areas*, which are aligned with CLIP video embeddings, isolating and explicitly modeling motion-related components. This design enhances the precision and controllability of dynamic feature capture, which is essential for video reconstruction tasks. To further enhance subject-agnostic applicability, we propose a disentanglement module to sep-

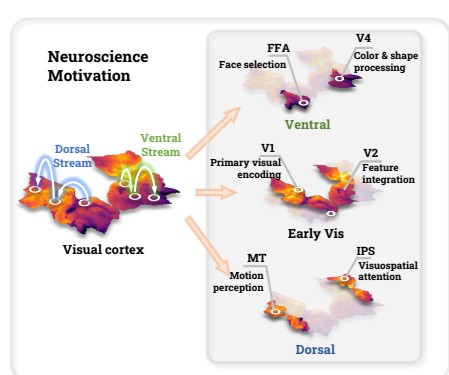

Figure 2: The visual cortex can be broadly divided into three types of areas: early visual, ventral, and dorsal. Early visual areas are primarily responsible for detecting low-level features including edges, orientation, and color. Ventral areas are associated with the processing of higher-level and abstract visual information. In contrast, dorsal areas are specialized for encoding dynamic features and spatial representations.

arate subject-specific and subject-agnostic semantic components. By employing contrastive learning techniques, we effectively extract robust semantic representations that are directly applicable to previously unseen subjects. (Fig. 1).

Our contributions can be summarized as follows: (1) We are the first to formulate fMRI-to-video decoding in a subject-agnostic setting, enabling the model to apply to previously unseen subjects **without any retraining**. (2) We propose VCFLOW, a novel subject-agnostic framework inspired by the ventral–dorsal dual-stream architecture of the visual cortex, which hierarchically extracts and aligns features in accordance with CLIP's cognitive hierarchy, reinforced by disentanglement

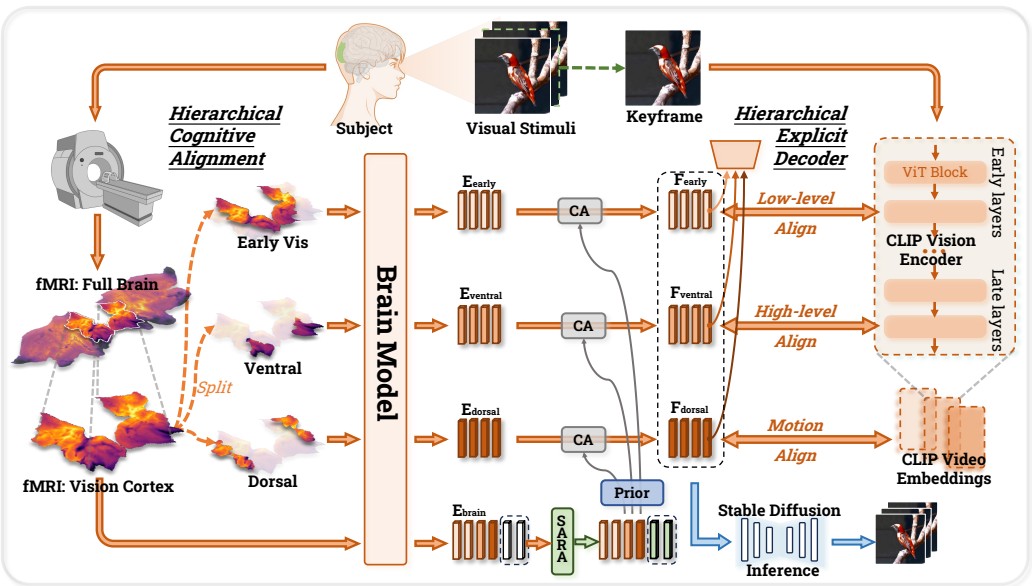

Figure 3: The overall framework of VCFLOW consists of three core components: (1) Hierarchical Cognitive Alignment Module (HCAM), (2) Subject-Agnostic Redistribution Adapter (SARA), and (3) Hierarchical Explicit Decoder (HED). VCFLOW learns three types of semantic representations through HCAM, which are then fused with subject-agnostic common features extracted by SARA. These enriched representations are subsequently decoded by HED to explicitly reconstruct information across multiple semantic levels.

strategy and feature-level contrastive learning. (3) Conventional subject-specific models require more than 12 hours of subject-specific data and substantial computational resources. In contrast, our subject-agnostic approach incurs only a **7% average drop** across all evaluation metrics compared to fully subject-specific approach while achieving **10-second inference** per video, making it fast, practical, and scalable for clinical use.

## 2 RELATED WORKS

### 2.1 FMRI-TO-VIDEO RECONSTRUCTION

In recent years, the field of fMRI signal reconstruction has witnessed rapid advancement, largely driven by the proliferation of fMRI technology and the concurrent development of deep learning architectures. Compared to the reconstruction of other modalities from fMRI data, the task of fMRI-to-video reconstruction presents significant challenges due to its comprehensive requirements for both semantic and temporal information. Despite these complexities, the fMRI-to-video reconstruction task holds considerable potential for understanding the cognitive processes underlying the perception of dynamic stimuli. Several notable studies have emerged recently to address this challenge (Chen et al., 2023; Gong et al., 2024; Wang et al., 2025; Lu et al.). For instance, MinD-Video (Chen et al., 2023) leverages diffusion models to generate videos with competitive semantic quality, yet it falls short in accurately capturing fine-grained visual details and maintaining consistent dynamic information across frames. NeuroClips (Gong et al., 2024) further mitigates this limitation by jointly leveraging keyframe reconstruction and temporally blurred video guidance. However, its robustness remains constrained, primarily due to an implicit alignment strategy with CLIP vision features. More recently, the NEURONS (Wang et al., 2025) approach attempts to integrate multidimensional information via the design of explicit training tasks. Nevertheless, critical information essential to these explicit tasks may not be sufficiently preserved within the diffusion prior formulation.

## 2.2 CROSS-SUBJECT LEARNING

In light of individual differences in brain structure and cognition, generalization across subjects typically yields significantly inferior performance compared to within-subject generalization. Nonetheless, cross-subject generalization remains essential, given its significance for real-time applications in clinical medicine and neuroscience (Wang et al., 2018; Sorger et al., 2012). In the fMRI-to-image domain, many studies have transitioned from individual-centric to holistic spatial methodologies (Scotti et al., 2024; Huo et al., 2024; Gong et al., 2025; Kong et al., 2024; Zhou et al., 2024); however, these approaches often still require subject-specific fine-tuning for previously unseen subjects. Similar trends have been observed within the video reconstruction domain (Li et al., 2024; Fosco et al., 2024). For instance, the GLFALi et al. (2024) method addresses this issue by functionally aligning data to a unified representational space. Nevertheless, such alignment strategies tend to lack semantic hierarchy and robustness, indicating room for further improvement in cross-subject generalization.

## 3 METHOD

The overall framework of VCFLOW is illustrated in Fig. 3. Inspired by the hierarchical cognitive processes of the human brain, VCFLOW integrates cross-subject semantic alignment at multiple cognitive levels. Specifically, VCFLOW comprises the following three modules: 1) **Hierarchical Cognitive Alignment Module (HCAM)**, which extracts cognitive features across hierarchical levels and aligns them within a unified semantic space(§3.1); 2) **Subject-Agnostic Redistribution Adapter (SARA)**, designed to map individual-specific semantic representations into a common, subject-invariant semantic space for robust cross-subject generalization(§3.2); and 3) **Hierarchical Explicit Decoder (HED)**, which decodes complementary features from different semantic dimensions and fuses them synergistically for accurate reconstruction(§3.3).

### 3.1 HIERARCHICAL COGNITIVE ALIGNMENT MODULE

#### 3.1.1 FUNCTIONAL ROI-BASED VOXEL PARTITIONING

Inspired by the dual-stream hypothesis of human visual processing, our cognitive workflow begins with early-stage perception, which is primarily associated with low-level information such as edges, color, orientation, and spatial structure. Subsequently, along the ventral stream, we select ROIs that capture higher-level semantics and associative memory. In addition, we select ROIs along the dorsal stream to enrich our representation with motion perception and spatial information (Kandel et al., 2000; Huff et al., 2018). Further discussion of the ROI selections is provided in the Appendix A. Let $\mathbf{X} \in \mathbb{R}^{B \times S \times V}$ denote the full fMRI voxel sequence, which is extracted from $\mathbf{X}_{\text{input}} \in \mathbb{R}^{B \times S \times H_{\text{input}} \times W_{\text{input}}}$, where $B$ is the batch size, $S$ is the number of subjects, and $V$ is the voxel length. Then,

$$\mathbf{X}_{\text{ROIs}} = \mathbf{X}[:, :, \mathcal{I}_{\text{ROIs}}] \in \mathbb{R}^{B \times S \times V_{\text{ROIs}}} \tag{1}$$

#### 3.1.2 MULTI-LEVEL FEATURE EXTRACTION

Due to the continuity and hierarchical nature of human cognition, directly extracting information from a subset of voxels for a specific dimension may disrupt the integrity of semantic representations. In contrast, a learning-based approach that integrates level-specific semantics with global representations provides a more coherent and interpretable modeling of brain activity.

Given the input fMRI signal $\mathbf{X}_{\text{input}}$, we extract four types of features: a full-brain representation $\mathbf{E}_{\text{brain}}$, early visual features $\mathbf{E}_{\text{early}}$, ventral stream features $\mathbf{E}_{\text{ventral}}$, and dorsal stream features $\mathbf{E}_{\text{dorsal}}$. The global representation $\mathbf{E}_{\text{brain}} \in \mathbb{R}^{B \times S \times D}$ is obtained by applying a ViT backbone to the entire voxel input $\mathbf{X}_{\text{input}}$. Meanwhile, the early visual subset $\mathbf{X}_{\text{early}}$, ventral stream subset $\mathbf{X}_{\text{ventral}}$, and dorsal stream subset $\mathbf{X}_{\text{dorsal}}$ are independently projected into the same latent feature space, resulting in $\mathbf{E}_{\text{early}}$, $\mathbf{E}_{\text{ventral}}$, and $\mathbf{E}_{\text{dorsal}}$, respectively. For the global representation $\mathbf{E}_{\text{brain}}$, after extracting universal features through SARA (Sec. 3.1), we follow prior work and employ an expressive diffusion prior to transform it into $\mathbf{F}_{\text{brain}} \in \mathbb{R}^{B \times S \times L_{\text{clip}} \times C_{\text{clip}}}$, optimized using the same loss $\mathcal{L}_{\text{prior}}$ as adopted in DALL·E 2 (Ramesh et al., 2022). This prior effectively facilitates the transformation

of features from the fMRI domain to the OpenCLIP embedding space, enabling subsequent reconstruction. Consequently, we utilize a learnable Cross-Attention module to guide the integration of features across different cognitive levels, yielding the representations $\mathbf{F}_{\text{early}}$, $\mathbf{F}_{\text{ventral}}$, and $\mathbf{F}_{\text{dorsal}}$.

### 3.1.3 HIERARCHICAL ALIGNMENT

To align multi-dimensional features within the OpenCLIP embedding space, we identify the most representative ground-truth features for each dimension. High-level features can be naturally aligned using CLIP vision embeddings $\mathbf{F}_{\text{clip}}^{(L)}$, given their rich semantic content. Aligning low-level information is relatively challenging; hence, we align these features with embeddings $\mathbf{F}_{\text{clip}}^{(l)}$ from early CLIP ViT layers, guided by neuroscientific insights into the hierarchical correspondence between deep neural networks and the human visual system (Yang et al., 2024). In the alignment process, we adopt the BiMixCo loss (Kim et al., 2020), which leverages MixCo-based data augmentation to construct a bidirectional contrastive objective that facilitates model convergence, detailed in the Appendix A.

### 3.2 SUBJECT-AGNOSTIC REDISTRIBUTION ADAPTER

### 3.2.1 REDISTRIBUTION LAYER

Inspired by previous ViT register-based feature extraction strategies (Darcet et al., 2023), we propose a unified semantic feature extraction framework based on a redistribution block, which serves as a token-level information classifier to separate and structure semantic information. To accommodate a more general scenario, we consider multi-subject feature sequences as the input to our framework. Let the input feature be defined as $\mathbf{E} \in \mathbb{R}^{B \times S \times L \times C}$, where $B$ is the batch size, $S$ is the number of subjects, $L$ denotes the number of temporal frames or patch tokens, and $C$ is the original feature dimension. To enhance the model's representational capacity, the feature is first expanded along the token dimension:

$$\mathbf{E}_{\text{exp}} = \text{Expand}(\mathbf{E}) \in \mathbb{R}^{B \times S \times (L+L_{\text{redis}}) \times C}. \tag{2}$$

The expanded representation $\mathbf{E}_{\text{exp}}$ is then passed through a redistribution layer to generate two distinct sets of latent tokens:

$$[\mathbf{T}_{\text{sem}}, \mathbf{T}_{\text{subj}}] = \text{Redistribution}(\mathbf{E}_{\text{exp}}), \tag{3}$$

where the semantic tokens $\mathbf{T}_{\text{sem}} \in \mathbb{R}^{B \times S \times L \times C}$ and the subject-specific tokens $\mathbf{T}_{\text{subj}} \in \mathbb{R}^{B \times S \times L_{\text{redis}} \times C}$ correspond to the original and expanded token groups, respectively. This redistribution mechanism enables the model to explicitly disentangle generalizable semantic content from subject-dependent features, thereby facilitating more robust and interpretable cross-subject alignment. In this task, $L$ and $C$ correspond to the token length and channel dimension of features extracted by OpenCLIP, respectively.

### 3.2.2 TRAINING OBJECTIVES

To achieve the above objectives, we define a set of training losses based on the semantic and subject tokens produced by the redistribution block.

First, we apply a BiMixCo-based alignment loss to encourage the semantic tokens to align with CLIP vision embeddings. Given semantic tokens $\mathbf{T}_{\text{sem}}$ and corresponding CLIP embeddings $\mathbf{F}_{\text{clip}}$, the semantic alignment loss is defined as:

$$\mathcal{L}_{\text{align}} = \text{BiMixCo}(\mathbf{T}_{\text{sem}}, \mathbf{F}_{\text{clip}}) \tag{4}$$

Building upon the semantic alignment, we further aim to project subject-specific semantics into a shared latent space that ensures semantic consistency across individuals. To achieve this, we adopt an inter-subject alignment strategy based on a bidirectional contrastive learning scheme. Specifically, we construct a moving window across subjects and apply a symmetric InfoNCE loss to enforce mutual alignment. Notably, as the number of subjects increases, this training paradigm becomes more effective and stable due to the richer inter-subject comparisons.

$$\mathcal{L}_{\text{generic}} = \frac{1}{2(S-1)} \sum_{i=2}^{S} \left[ \text{InfoNCE}\left(\mathbf{T}_{i-1,\text{sem}}^{\text{norm}}, \mathbf{T}_{i,\text{sem}}^{\text{norm}}\right) + \text{InfoNCE}\left(\mathbf{T}_{i,\text{sem}}^{\text{norm}}, \mathbf{T}_{i-1,\text{sem}}^{\text{norm}}\right) \right] \tag{5}$$

To complement the semantic alignment, it is essential to preserve subject-specific identity information embedded within the individualized semantic deviations as well. To this end, we introduce a subject classifier trained on $\mathbf{T}_{\text{subj}}$, where each token, along the subject dimension, corresponds to a subject-specific representation. The classifier is supervised using a cross-entropy loss that encourages the retention of discriminative individual features, where $y_{\text{subj}}^{(k)}$ denotes the one-hot ground-truth label for subject $k$ and $z^{(k)}$ represents the classifier's logit score corresponding to the prediction that the input belongs to subject $k$:

$$\mathcal{L}_{\text{subj}} = -\sum_{k=1}^{K} y_{\text{subj}}^{(k)} \log \left( \frac{\exp(z^{(k)})}{\sum_{j=1}^{K} \exp(z^{(j)})} \right) \tag{6}$$

The total loss function is a weighted combination of all objectives:

$$\mathcal{L}_{\text{SARA}} = \lambda_{\text{align}} \mathcal{L}_{\text{align}} + \lambda_{\text{subj}} \mathcal{L}_{\text{subj}} + \lambda_{\text{generic}} \mathcal{L}_{\text{generic}} \tag{7}$$

### 3.3 HIERARCHICAL EXPLICIT DECODER

In the reconstruction of videos, directly utilizing extracted feature embeddings may fail to adequately integrate information across multiple cognitive levels. However, decoding these embeddings into existing auxiliary modalities—such as textual descriptions, segmentation masks, and blurry video representations—can significantly enhance reconstruction quality. This conversion is effectively achieved by formulating explicit auxiliary tasks. Within the **Hierarchical Cognitive Alignment Module**, we systematically generate three distinct features: $\mathbf{F}_{\text{early}}$, $\mathbf{F}_{\text{ventral}}$, and $\mathbf{F}_{\text{dorsal}}$, each associated with tailored explicit tasks that refine feature representations within their respective cognitive dimensions, thus improving reconstruction performance (Fig. 4).

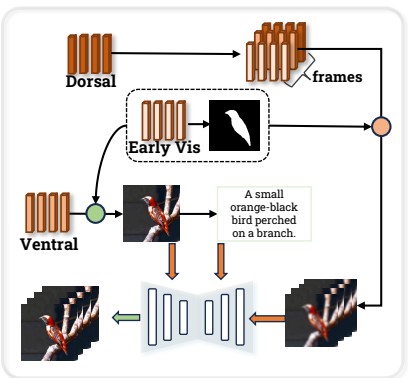

Figure 4: The inference stage of VCFLOW integrates multi-level semantic embeddings to facilitate comprehensive decoding.

For the ventral stream feature $\mathbf{F}_{\text{ventral}}$, which encodes abstract semantic content such as object identity and categorical meaning, we introduce two explicit tasks: image caption generation and object category classification, yielding the losses $\mathcal{L}_{\text{caption}}$ and $\mathcal{L}_{\text{cls}}$, respectively. For the early visual feature $\mathbf{F}_{\text{early}}$, which represents perceptual and structural properties such as edges, textures, and spatial layout, we design a segmentation task to effectively capture morphological details, resulting in the segmentation loss $\mathcal{L}_{\text{seg}}$. Regarding the dorsal stream feature $\mathbf{F}_{\text{dorsal}}$, which characterizes spatial-temporal dynamics and motion-related cues, we align it with blurry video modalities to explicitly capture spatial-temporal motion information. Specifically, we first project $\mathbf{F}_{\text{dorsal}}$ into the video frame dimension, obtaining $\widetilde{\mathbf{F}}_{\text{dorsal}} \in \mathbb{R}^{B \times F \times S \times L_{\text{clip}} \times C_{\text{clip}}}$, where $F$ represents the frame dimension. This transformed feature is further projected into the latent space of a Variational Autoencoder (VAE) to achieve alignment, yielding the alignment loss $\mathcal{L}_{\text{motion}}$.

Finally, the total loss is formulated as:

$$\mathcal{L}_{\text{HED}} = \lambda_{\text{caption}} \mathcal{L}_{\text{caption}} + \lambda_{\text{cls}} \mathcal{L}_{\text{cls}} + \lambda_{\text{seg}} \mathcal{L}_{\text{seg}} + \lambda_{\text{motion}} \mathcal{L}_{\text{motion}} \tag{8}$$

Following NEURONS (Wang et al., 2025), we progressively adjust the loss coefficients during training, as shown in Appendix A.

## 4 EXPERIMENT

### 4.1 FMRI-IMAGE DATASETS AND PRETRAINING

We pretrain the backbone on the DIR dataset (Shen et al., 2019) and the GOD dataset (Horikawa & Kamitani, 2017) to acquire an initial capacity for capturing semantic representations from neural

activity. In the DIR and GOD datasets, fMRI signals were recorded from eight subjects while they viewed 1,250 natural images spanning 200 categories using a 3.0-Tesla Siemens MAGNETOM Verio scanner. The visual stimuli used in the image presentation experiments for both datasets were identical and sourced from ImageNet (Deng et al., 2009). Among these images, 1,200 images from 150 categories were allocated to the training sessions, whereas 50 images from 50 categories were reserved for the test sessions.

Due to the lack of dynamic information in image datasets, the primary goal of our pretraining stage is to train the backbone to acquire an initial understanding of semantics and to develop the ability to separate subject tokens. Therefore, we adopt the SARA training objective, as shown in Equation 7.

## 4.2 Dataset and Preprocessing

In this study, we conducted fMRI-to-video reconstruction experiments using the publicly available fMRI-video dataset, cc2017 dataset (Wen et al., 2018). For each subject, 18 training video clips and 5 testing video clips—each 8 minutes in duration—were presented. The training clips were shown twice, while the testing clips were presented 10 times, and the corresponding fMRI signals in the test set were averaged across repetitions to improve signal-to-noise ratio. MRI data (including T1- and T2-weighted anatomical scans) and fMRI data (with a temporal resolution of 2 seconds) were acquired using a 3-Tesla MRI scanner. In total, the dataset contains 8640 training samples and 1200 testing samples of synchronized fMRI-video pairs per subject.

For fMRI data preprocessing, we follow the strategy proposed in fMRI-PTE (Qian et al., 2023) to map fMRI signals into a common representational space, upon which brain regions are further segmented. Considering the hemodynamic response delay between stimulus onset and the peak of the BOLD signal, we introduce a temporal shift of approximately 6 seconds to the fMRI data. Further details are provided in the Appendix B.

## 4.3 Evaluation Metrics

To comprehensively evaluate the reconstruction quality, we adopt a two-fold assessment strategy, considering both frame-level and video-level performance. At the frame level, we evaluate both semantic consistency and pixel-wise similarity. For semantic evaluation, we perform an N-way top-K classification task based on 1,000 ImageNet categories. Each trial compares the classification result of a predicted frame against its ground truth counterpart. A trial is deemed correct if the ground truth label appears within the top-K predictions (top-1 in our case) of the predicted frame, randomly selected from N candidate labels. The final accuracy is averaged over 100 repeated trials. For pixel-level assessment, we report SSIM and PSNR scores to capture structural and intensity-level differences between the reconstructed and original frames. At the video level, we examine two key aspects: semantic accuracy and spatiotemporal (ST) continuity. Semantic accuracy is also measured through an N-way top-K action classification task (top 1 in our case) involving 400 classes from the Kinetics-400 dataset (Kay et al., 2017), using a VideoMAE-based model (Tong et al., 2022) as the classifier. To evaluate spatiotemporal coherence, we calculate the CLIP embedding for each video frame and compute the mean cosine similarity between every pair of adjacent frames. This metric, commonly referred to as CLIP-pcc in the video editing literature (Wu et al., 2023), reflects how smoothly semantic content transitions over time.

## 4.4 Main Results

### 4.4.1 Quantitative Results

We evaluated the performance of VCFLOW against existing methods, conducting comparisons across three subjects as well as their averaged results. As shown in Table 1, our approach achieves substantial improvements over the subject-agnostic baselines NEURONS$^*$ and GLFA$^*$, and even surpasses GLFA, which pretrains the encoder on fMRI data from all subjects. These results highlight the superiority of our framework.

To be specific, the frame-based metrics exhibit substantial improvements. At the semantic level, our method attains an accuracy of 14.0% on the 50-way classification task, representing a **46% relative gain** over GLFA$^*$ (9.6%). Consistent improvements are also observed in pixel-level metrics,

| Task | Method | w/o Pretrain | Frame-based | | | | Video-based | | |
|---|---|---|---|---|---|---|---|---|---|
| | | | Semantic-level | | Pixel-level | | Semantic-level | | ST-level |
| | | | 50-way↑ | 2-way↑ | SSIM↑ | PSNR↑ | 50-way↑ | 2-way↑ | CLIP-pcc↑ |
| 2,3 → 1 | fMRI-PTE-V (Li et al., 2024) | × | 12.3% | 77.1% | 0.151 | - | 17.9% | 84.6% | - |
| | GLFA (Li et al., 2024) | × | 12.6% | 78.1% | 0.181 | - | 17.9% | 83.7% | - |
| | NEURONS* (Wang et al., 2025) | ✓ | 9.7% | 74.3% | 0.377 | 9.200 | 15.5% | 82.9% | 0.926 |
| | GLFA* (Li et al., 2024) | ✓ | 9.0% | 74.1% | 0.133 | - | 16.3% | 83.3% | - |
| | VCFLOW | ✓ | **14.2%** | **78.6%** | **0.389** | **10.469** | **18.9%** | **84.8%** | **0.944** |
| 1,3 → 2 | fMRI-PTE-V (Li et al., 2024) | × | 10.4% | 76.2% | 0.130 | - | 17.4% | 84.4% | - |
| | GLFA (Li et al., 2024) | × | 10.5% | 76.8% | 0.167 | - | 17.5% | 83.8% | - |
| | NEURONS* (Wang et al., 2025) | ✓ | 10.6% | 75.5% | 0.385 | 10.000 | 16.8% | 84.3% | 0.936 |
| | GLFA* (Li et al., 2024) | ✓ | 10.3% | 74.9% | 0.141 | - | 17.7% | **84.6%** | - |
| | VCFLOW | ✓ | **13.2%** | **77.6%** | **0.424** | **10.629** | **18.0%** | 84.3% | **0.937** |
| 1,2 → 3 | fMRI-PTE-V (Li et al., 2024) | × | 10.7% | 76.5% | 0.161 | - | 18.2% | 83.4% | - |
| | GLFA (Li et al., 2024) | × | 11.6% | **77.7%** | 0.172 | - | **19.3%** | **84.7%** | - |
| | NEURONS* (Wang et al., 2025) | ✓ | 10.0% | 74.9% | **0.378** | 9.636 | 16.0% | 83.6% | 0.931 |
| | GLFA* (Li et al., 2024) | ✓ | 9.5% | 75.3% | 0.137 | - | 17.0% | 84.1% | - |
| | VCFLOW | ✓ | **14.7%** | 77.6% | 0.375 | **10.335** | 17.8% | 84.5% | **0.940** |
| Average | fMRI-PTE-V (Li et al., 2024) | × | 11.1% | 76.6% | 0.147 | - | 17.8% | 84.1% | - |
| | GLFA (Li et al., 2024) | × | 11.6% | 77.5% | 0.173 | - | 18.2% | 84.1% | - |
| | NEURONS* (Wang et al., 2025) | ✓ | 10.1% | 74.9% | 0.380 | 9.612 | 16.1% | 83.6% | 0.931 |
| | GLFA* (Li et al., 2024) | ✓ | 9.6% | 74.8% | 0.137 | - | 17.0% | 84.0% | - |
| | VCFLOW | ✓ | **14.0%** | **77.9%** | **0.396** | **10.478** | **18.2%** | **84.5%** | **0.940** |
| Comp. | VCFLOW vs. GLFA* (Δ%) | - | +45.8% | +4.1% | +189.1% | - | +7.1% | +0.6% | - |
| | VCFLOW vs. NEURONS* (Δ%) | - | +38.6% | +4.0% | +4.2% | +9.0% | +13.0% | +1.1% | +1.0% |
| | VCFLOW vs. GLFA (Δ%) | - | +20.7% | +0.5% | +128.9% | - | 0.0% | +0.5% | - |

Table 1: Quantitative comparison of VCFLOW with representative methods. All results are based on subjects provided by the cc2017 dataset (Wen et al., 2018). GLFA* refers to GLFA results with test subject data excluded during pretraining. NEURONS* refers to the NEURONS model adapted to a subject-agnostic setting by modifying its data processing pipeline and encoder components. *w/o Pretrain* indicates whether the encoder is pretrained using the fMRI data of the **test subject**.

including SSIM and PSNR. Moreover, the video-based metrics achieve state-of-the-art performance under the current setting, highlighting the model's superior ability to capture dynamic information. These results collectively underscore the effectiveness of VCFLOW in both motion modeling and semantic feature extraction. The results comparing with subject-specific methods are provided in Appendix C.

### 4.4.2 QUALITATIVE RESULTS

We compare the qualitative performance of VCFLOW with GLFA, as illustrated in Fig. 5. The results demonstrate that our method achieves superior performance in both semantic accuracy and the ability to capture motion dynamics. Benefiting from the incorporation of dedicated motion features, the reconstructed videos exhibit more coherent and faithful representations of spatial and temporal information. Additional results are provided in the Appendix C.

### 4.5 ABLATION STUDIES

In this section, we evaluate the effectiveness of four key components: fMRI-to-image pretraining, HCAM, SARA, and HED. All experiments follow a subject-agnostic setting, where models are trained on subjects 2 and 3 and directly tested on subject 1. As shown in Table 2, each module contributes to performance gains across both video-based and image-based metrics. Adding HCAM enhances semantic understanding, while SARA boosts cross-subject transferability, reflected in CLIP-pcc and PSNR. Introducing HED provides the most significant gains, particularly in high-level semantics and reconstruction quality.

### 4.6 INTERPRETATION RESULTS

By adopting the cortical projection technique introduced in (Yang et al., 2024), we visualize the hierarchical embeddings on the brain surface map. As illustrated in Fig. 6, the early vision embeddings exhibit a clear correspondence with early visual areas, notably V1 through V4. The ventral stream embeddings show strong activation in high-level visual regions such as the FFA and PPA, while also

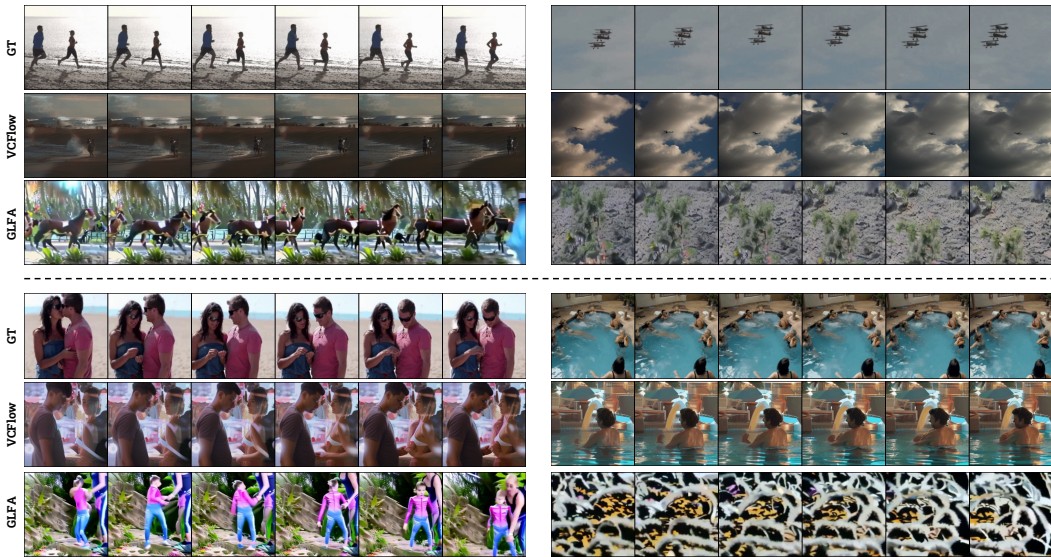

Figure 5: Compared with GLFA (Li et al., 2024), the qualitative comparison demonstrates that VCFLOW achieves superior semantic fidelity and temporal coherence, effectively capturing fine-grained semantics and preserving motion information in a subject-agnostic setting.

| Brain | Image | Modules | | | Frame-based | | | | Video-based | | |
|---|---|---|---|---|---|---|---|---|---|---|---|
| Model | Pretrain | HCAM | SARA | HED | 50-way↑ | 2-way↑ | SSIM↑ | PSNR↑ | 50-way↑ | 2-way↑ | CLIP-pcc↑ |
| ✓ | | | | | 11.3% | 73.1% | **0.401** | 9.720 | 12.6% | 81.3% | 0.908 |
| ✓ | ✓ | | | | 10.4% | 75.0% | 0.382 | 9.866 | 15.3% | 81.8% | 0.918 |
| ✓ | ✓ | ✓ | | | 11.8% | 75.9% | 0.357 | 9.583 | 14.7% | 82.4% | 0.919 |
| ✓ | ✓ | ✓ | ✓ | | 12.4% | 77.2% | 0.389 | 10.442 | 15.2% | 83.1% | 0.934 |
| ✓ | ✓ | ✓ | ✓ | ✓ | **14.2%** | **78.6%** | 0.389 | **10.469** | **18.9%** | **84.8%** | **0.944** |

Table 2: Ablations on the key components of VCFLOW, and all results are from subject 1.

demonstrating more diffuse projections across broader cortical areas, likely due to their encoding of abstract global semantics. In contrast, the dorsal stream embeddings align well with motion-related regions, particularly those associated with MST. These interpretation results are highly consistent with the neurocognitive structure underpinning our proposed framework.

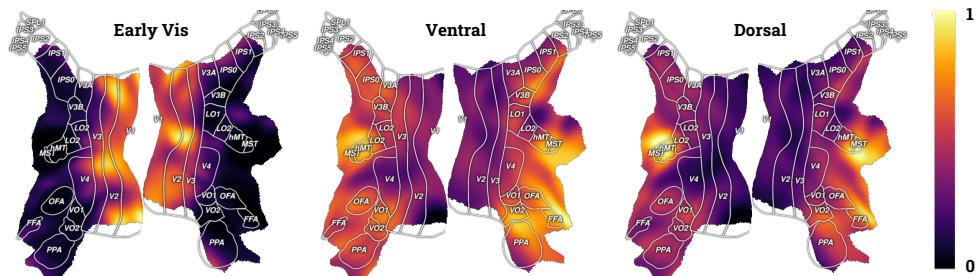

Figure 6: By projecting the three types of embeddings from different semantic dimensions onto the cortical surface, we can clearly observe their respective correspondences with different brain regions.

## 5 CONCLUSION

In this work, we propose VCFLOW, the first subject-agnostic framework for fMRI-to-video reconstruction. We design a novel dual-stream hierarchical feature extraction architecture inspired by the

human brain's cognitive pathways, enabling the model to extract multi-level features for accurate video reconstruction. A key innovation of our approach lies in the utilization of CLIP embeddings from multiple layers to achieve fine-grained semantic alignment with fMRI signals. To further enhance generalization across individuals, we introduce a redistribution-based cross-subject learning strategy that captures subject-invariant representations. Without relying on any subject-specific training data, VCFLOW achieves efficient video reconstruction with only a marginal drop in accuracy, offering a rapid and generalizable solution suitable for clinical applications. Additionally, our interpretation analyses provide compelling evidence of the alignment between the reconstructed features and human visual cognition mechanisms.

## ACKNOWLEDGEMENTS

This work was supported in part by the Joint Research Scheme (JRS) under the National Natural Science Foundation of China (NSFC) and the Research Grants Council (RGC) of Hong Kong (Project No. N_HKUST654/24); the Research Grants Council of the Hong Kong Special Administrative Region, China (Project Reference No. AoE/E-601/24-N and Project No. R6005-24); and the National Natural Science Foundation of China (Grant No. 62306254).

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

# A  TECHNICAL DETAILS

## A.1  MIXCO LOSS

The MixCo needs to mix two independent fMRI signals. For each $\mathcal{Y}_c$, we random sample another fMRI $\mathcal{Y}_{m_c}$, which is the keyframe of the clip index by $m_c$. Then, we mix $\mathcal{Y}_c$ and $\mathcal{Y}_{m_c}$ using a linear combination:

$$\mathcal{Y}_c^* = mix(\mathcal{Y}_c, \mathcal{Y}_{m_c}) = \lambda_c \cdot \mathcal{Y}_c + (1 - \lambda_c)\mathcal{Y}_{m_c}, \tag{9}$$

where $\mathcal{Y}_c^*$ denotes mixed fMRI signal and $\lambda_c$ is a hyper-parameter sampled from Beta distribution. Then, we adapt the ridge regression to map $\mathcal{Y}_c^*$ to a lower-dimensional $\mathcal{Y}_c^{*'}$ and obtain the embedding $e_{\mathcal{Y}_c^*}$ via the MLP, i.e. $e_{\mathcal{Y}_c^*} = \mathcal{E}(\mathcal{Y}_c^{*'})$. Based on this, the BiMixCo loss can be formed as:

$$
\begin{aligned}
\mathcal{L}_{\text{BiMixCo}} = &-\frac{1}{2N_f} \sum_{i=1}^{N_f} \lambda_i \cdot \log \frac{\exp\big(\text{sim}(e_{\mathcal{Y}_i^*}, e_{\mathcal{X}_i})/\tau\big)}{\sum_{k=1}^{N_f} \exp\big(\text{sim}(e_{\mathcal{Y}_i^*}, e_{\mathcal{X}_k})/\tau\big)} \\
&-\frac{1}{2N_f} \sum_{i=1}^{N_f} (1 - \lambda_i) \cdot \log \frac{\exp\big(\text{sim}(e_{\mathcal{Y}_i^*}, e_{\mathcal{X}_{m_i}})/\tau\big)}{\sum_{k=1}^{N_f} \exp\big(\text{sim}(e_{\mathcal{Y}_i^*}, e_{\mathcal{X}_k})/\tau\big)} \\
&-\frac{1}{2N_f} \sum_{j=1}^{N_f} \lambda_j \cdot \log \frac{\exp\big(\text{sim}(e_{\mathcal{Y}_j^*}, e_{\mathcal{X}_j})/\tau\big)}{\sum_{k=1}^{N_f} \exp\big(\text{sim}(e_{\mathcal{Y}_k^*}, e_{\mathcal{X}_j})/\tau\big)} \\
&-\frac{1}{2N_f} \sum_{j=1}^{N_f} \sum_{\{l|m_l=j\}} (1 - \lambda_j) \cdot \log \frac{\exp\big(\text{sim}(e_{\mathcal{Y}_l^*}, e_{\mathcal{X}_j})/\tau\big)}{\sum_{k=1}^{N_f} \exp\big(\text{sim}(e_{\mathcal{Y}_k^*}, e_{\mathcal{X}_j})/\tau\big)}
\end{aligned}
\tag{10}
$$

where $e_{\mathcal{X}_c}$ denotes the OpenCLIP embedding for keyframe $\mathcal{X}_c$.

### A.2 PRIOR LOSS

We use the Diffusion Prior to transform fMRI embedding $e_{\mathcal{Y}_c}$ into the reconstructed OpenCLIP embeddings of video $e_{\mathcal{V}_c}$. Similar to DALLE·2, Diffusion Prior predicts the target embeddings with mean-squared error (MSE) as the supervised objective:

$$\mathcal{L}_{\text{Prior}} = \mathbb{E}_{e_{\mathcal{V}_c}, e_{\mathcal{Y}_c}, \epsilon \sim \mathcal{N}(0,1)} ||\epsilon(e_{\mathcal{Y}_c}) - e_{\mathcal{V}_c}||. \tag{11}$$

### A.3 FUNCTIONAL CLASSIFICATION OF ROIS

In constructing the dual-stream framework, we distinguish three main groups: early vision ROIs, dorsal ROIs, and ventral ROIs. However, the dual-stream theory does not provide a definitive assignment for every ROI in the visual cortex, and the categorization of certain borderline regions remains ambiguous. To address this uncertainty, we adopted two alternative ROI partitioning schemes and compared their effectiveness. Considering both cognitive relevance and the balance in voxel distribution, we ultimately selected scheme A as our final partitioning strategy. The comparative results on subject 1 are presented in Table 3.

**Scheme A**

- Early vision: V1, V2, V3, V4
- Dorsal stream: V3A, V3B, V6, V6A, V7, IPS1, LO1, LO2, LO3, FST, MT, MST, V3CD, V4t, PH, IP0
- Ventral stream: FFC, PIT, V8, VMV1, VMV2, VMV3, VVC, PHA1, PHA2, PHA3, TE2p

**Scheme B**

- Early vision: V1, V2, V3, V4
- Dorsal stream: V3A, V3B, V6, V6A, V7, IPS1, FST, MT, MST, V3CD, V4t, IP0
- Ventral stream: FFC, PIT, V8, VMV1, VMV2, VMV3, VVC, PHA1, PHA2, PHA3, TE2p, LO1, LO2, LO3, PH

| Scheme | Frame-based | | | | Video-based | | |
|---|---|---|---|---|---|---|---|
| | Semantic-level | | Pixel-level | | Semantic-level | | ST-level |
| | 50-way↑ | 2-way↑ | SSIM↑ | PSNR↑ | 50-way↑ | 2-way↑ | CLIP-pcc↑ |
| Scheme A | **14.2%** | **78.6%** | **0.389** | **10.469** | **18.9%** | **84.8%** | **0.944** |
| Scheme B | 12.4% | 71.9% | 0.353 | 9.366 | 9.8% | 80.6% | 0.913 |

Table 3: Comparison of ROI partitioning schemes on subject 1.

### A.4 PROGRESSIVE LEARNING STRATEGY

We adopt a progressive learning strategy (Wang et al., 2025) to jointly optimize multiple loss functions. This strategy regulates the weight of each loss along a smooth schedule: starting from 1, increasing to 10, and then decaying back to 1 within a predefined period. The scheduling is applied to four loss functions, each with a distinct offset period to promote balanced training.

Formally, let $E$ denote the epoch index, $B$ the batch index, and $N_B$ the number of batches per epoch. The total number of batches within a period $P$ is defined as $T = P \cdot N_B$. For a period that starts at epoch $S$, the position of the current batch within this period is computed as

$$C = (E - S) \cdot N_B + B. \tag{12}$$

The weight $w$ at this position is then given by

$$w = 1 + 9 \cdot \left| \sin\left( \frac{C}{T} \cdot \pi \right) \right|. \tag{13}$$

If $E$ lies outside the interval $[S, S + P)$, the weight remains constant, i.e., $w = 1$.

## A.5 SUBTASKS DESCRIPTIONS

Our task setup is inspired by NEURONS (Wang et al., 2025), and we describe each subtask in detail below. We first present the ventral-related tasks, which are mainly designed to capture high-level semantic information along the visual processing stream.

**Concept Recognition.** To enhance conceptual understanding, we introduce a concept recognition task by adding a multi-label classifier $\mathcal{D}_{cls}(\cdot)$ that predicts the key concepts in each frame from the fMRI-derived visual embeddings. Concretely, we apply a cross-entropy loss between the classifier prediction and the ground-truth (GT) concept list:

$$\mathcal{L}_{cls} = \mathcal{L}_{ce}(\mathcal{D}_{cls}(\bar{e}^v), \mathcal{C}), \tag{14}$$

where $\bar{e}^v$ denotes the mean of $e^v$ along the frame axis, and $\mathcal{C}$ is the GT concept list.

**Scene Description.** To further model scene-level semantics, we incorporate a scene description task that aims to generate a descriptive caption for each video frame. Specifically, we finetune a pre-trained text decoder $\mathcal{D}_{caption}(\cdot)$, which takes the fMRI-derived text embeddings $e^t$ as input and produces the caption. We adopt GPT-2 (Radford et al., 2019) as the text decoder and train it using prefix language modeling. Given a GT caption token sequence $\mathcal{S} = \{s_0, s_1, \ldots, s_{|\mathcal{S}|}\}$ and the corresponding text embedding $e^t$, the decoder $\mathcal{D}_{caption}(\cdot)$ is trained to reconstruct $\mathcal{S}$ conditioned on $e^t$ as the prefix. The training objective $\mathcal{L}_{caption}$ is defined as:

$$\mathcal{L}_{caption} = -\frac{1}{|\mathcal{S}|} \sum_{i=1}^{|\mathcal{S}|} \log \mathcal{D}_{caption}(s_i \mid s_{<i}, e^t), \tag{15}$$

where $s_i$ denotes the $i$-th token in the GT sequence $\mathcal{S}$.

Next, we consider early-vision tasks, which primarily aim to learn coarse object contours and spatial masks.

**Key Object Segmentation.** To better capture object-level information, we design a text-driven video decoder based on the VAE video decoder (von Platen et al., 2022). This decoder takes the video embeddings $e^v$ together with the text embeddings $e^t$ as inputs. For this task, the text embeddings are obtained by encoding the category name of the key object with the CLIP text encoder. A cross-attention module is then applied to activate specific patches in $e^v$ (used as queries $Q$) corresponding to $e^t$ (used as keys $K$ and values $V$): $e^{seg} = \text{softmax}\left(\frac{QK^\top}{\sqrt{d}}\right) \cdot V$. The activated feature $e^{seg}$ is upsampled to a higher resolution for pixel-level prediction. We then employ a simple segmentation head $\mathcal{D}_{vs}(\cdot)$ to generate the binary segmentation masks $y_{seg}$ for the key object in the video. The training objective for this task is a binary cross-entropy loss, denoted as $\mathcal{L}_{seg}$.

Finally, we introduce the dorsal-related task, which focuses on modeling global motion information.

**Blurry Video Reconstruction.** We reuse the same VAE decoder as in the key object segmentation task, but replace the segmentation head $\mathcal{D}_{vs}(\cdot)$ with a reconstruction head $\mathcal{D}_{vr}(\cdot)$. Given the video embeddings $e^v$, the text-driven video decoder together with $\mathcal{D}_{vr}(\cdot)$ generates a blurry video $y_c^{motion}$. We then map $y_c^{motion}$ into the latent space of the Stable Diffusion VAE to obtain the latent embeddings $y_c'$. This subtask is trained with a mean absolute error (MAE) loss, defined as:

$$\mathcal{L}_{motion} = \frac{1}{F} \sum_{i=1}^{F} \left| y_{c,i}^{motion} - y_{c,i}' \right|. \tag{16}$$

It is worth noting that, unlike NEURONS, which jointly optimizes all tasks over a single global feature representation, we explicitly associate each task with a particular cognitive process and train it on the corresponding process-specific features.

## A.6 INFERENCE

At inference time, our pipeline leverages a pre-trained T2V diffusion model (Guo et al., 2023), in a setup similar to NEURONS (Wang et al., 2025). We conditions the model jointly on a control image, a blurry video, and a text description. These three inputs are assembled from the outputs of

the decoupled tasks: a control image is reconstructed from each frame via unCLIP (Ramesh et al., 2022), while $\mathcal{D}_{cls}$ and $\mathcal{D}_{caption}$ provide the predicted concepts and generated captions, respectively. The embedding of the top-1 predicted concept, together with the caption embedding, is used to guide video-mask prediction through $\mathcal{D}_{vs}$ and blurry video reconstruction through $\mathcal{D}_{vr}$. To further enhance the prominence of the key object, we rescale its binary mask from $0, 1$ to $[0.5, 1]$ and multiply it with both the control image and the blurry video before feeding them into the diffusion model.

Because inference for unseen subjects does not require any subject-specific finetuning, their data can be directly processed by the model. With this setup, the inference time—normalized by the total number of videos—remains below 10 seconds per video. All measurements are obtained on an NVIDIA RTX 4090 GPU.

### A.7 DETAILS OF BRAIN MODEL

As shown in Fig. 3, our Brain Model extracts four types of representations from the input fMRI signals: a full-brain representation $\mathbf{E}_{\text{brain}}$, early visual features $\mathbf{E}_{\text{early}}$, ventral-stream features $\mathbf{E}_{\text{ventral}}$, and dorsal-stream features $\mathbf{E}_{\text{dorsal}}$. Here, the Brain Model refers to the module responsible for deriving these fMRI-based features, consisting of a ViT backbone together with several linear projection heads.

The linear heads operate directly on the flattened voxel signals to obtain the three cognitively grounded feature sets—$\mathbf{E}_{\text{early}}$, $\mathbf{E}_{\text{ventral}}$, and $\mathbf{E}_{\text{dorsal}}$—each with shape $\mathbb{R}^{B \times S' \times D}$. For the global full-brain representation $\mathbf{E}_{\text{brain}}$, we adopt a ViT-based fMRI encoder (fMRI-PTE (Qian et al., 2023), pre-trained on the UK Biobank dataset (Miller et al., 2016)), which produces features of shape $\mathbb{R}^{B \times S \times D}$.

## B DETAILS OF DATA PRE-PROCESSING

We employ both the fMRI-to-image datasets and the fMRI-to-video dataset in our experiments (Wen et al., 2018; Shen et al., 2019; Horikawa & Kamitani, 2017). For fMRI data preprocessing, we first aligned all fMRI volumes to the *32k_fs_LR* brain surface space based on anatomical structure (Glasser et al., 2013). Unlike conventional fMRI decoding methods that flatten each fMRI frame into a one-dimensional signal and select subject-specific activated voxels, our approach transforms the fMRI data into a unified surface-based representation across subjects. After surface alignment, we applied voxel-wise $z$-transformation to normalize the fMRI signals and unfolded the cortical surface into a two-dimensional plane, thereby preserving spatial relationships between adjacent voxels. Given that only a subset of brain regions is typically activated during visual stimulation tasks (Huang et al., 2021), we further restricted the analysis to early and higher visual cortical Regions of Interest (ROIs). These ROIs, encompassing a total of 8,405 vertices, were defined according to the HCP-MMP atlas (Glasser et al., 2016) in the *32k_fs_LR* surface space. Each processed fMRI volume was then converted into a single-channel $256 \times 256$ image. For datasets with multiple runs corresponding to the same video stimulus, we averaged the aligned fMRI frames across runs. Finally, for fMRO-to-Video dataset, to account for the hemodynamic response delay—the time lag between stimulus presentation and the peak of the BOLD response—a temporal shift of approximately 6 seconds was applied to the fMRI data.

## C ADDITIONAL EXPERIMENTAL RESULTS

### C.1 COMPARISON WITH SUBJECT-SPECIFIC METHODS

We also compare our method with state-of-the-art approaches under different settings as shown in Table 4. Notably, when compared with the subject-specific state-of-the-art method NEU-RONS (Wang et al., 2025), our approach exhibits only a modest decrease of **7%** in average performance. However, it achieves the advantage of direct testing without the need for retraining.

### C.2 VISUALIZATION RESULTS

We further conduct a qualitative comparison among VCFLOW, GLFA (Li et al., 2024), and NEU-RONS (Wang et al., 2025), as illustrated in Fig. 7. Compared to GLFA, VCFLOW consistently

| Method | Setting | Frame-based | | | | Video-based | | |
|--------|---------|-------------|--|--|--|-------------|--|--|
| | | Semantic-level | | Pixel-level | | Semantic-level | | ST-level |
| | | 50-way↑ | 2-way↑ | SSIM↑ | PSNR↑ | 50-way↑ | 2-way↑ | CLIP-pcc↑ |
| *Subject 1* | | | | | | | | |
| NEURONS (Wang et al., 2025) | Subject-specific | 20.6% | 81.0% | 0.373 | 9.591 | 25.4% | 86.2% | 0.932 |
| GLFA (Li et al., 2024) | Subject-adaptive pretraining | 11.6% | 77.7% | 0.172 | - | 19.3% | 84.7% | - |
| VCFLOW | Subject-agnostic | 14.2% | 78.6% | 0.389 | 10.469 | 18.9% | 84.8% | 0.944 |
| *Subject 2* | | | | | | | | |
| NEURONS (Wang et al., 2025) | Subject-specific | 21.4% | 81.0% | 0.353 | 9.502 | 25.2% | 86.0% | 0.933 |
| GLFA (Li et al., 2024) | Subject-adaptive pretraining | 10.5% | 76.8% | 0.167 | - | 17.5% | 83.8% | - |
| VCFLOW | Subject-agnostic | 13.2% | 77.6% | 0.424 | 10.629 | 18.0% | 84.3% | 0.937 |
| *Subject 3* | | | | | | | | |
| NEURONS (Wang et al., 2025) | Subject-specific | 21.0% | 81.7% | 0.369 | 9.488 | 27.8% | 86.8% | 0.937 |
| GLFA (Li et al., 2024) | Subject-adaptive pretraining | 12.6% | 78.1% | 0.181 | - | 17.9% | 83.7% | - |
| VCFLOW | Subject-agnostic | 14.7% | 77.6% | 0.375 | 10.335 | 17.8% | 84.5% | 0.940 |
| *Average* | | | | | | | | |
| NEURONS (Wang et al., 2025) | Subject-specific | 21.0% | 81.2% | 0.365 | 9.527 | 26.1% | 86.3% | 0.934 |
| GLFA (Li et al., 2024) | Subject-adaptive pretraining | 11.6% | 77.5% | 0.173 | - | 18.2% | 84.1% | - |
| VCFLOW | Subject-agnostic | 14.0% | 77.9% | 0.396 | 10.478 | 18.2% | 84.5% | 0.940 |
| VCFLOW vs. GLFA (Δ%) | - | +20.7% | +0.5% | +128.9% | - | 0.0% | +0.5% | - |
| VCFLOW vs. NEURONS (Δ%) | - | -33.3% | -4.1% | +8.5% | +10.0% | -30.3% | -2.1% | +0.6% |
| GLFA vs. NEURONS (Δ%) | - | -44.8% | -4.6% | -52.6% | - | -30.3% | -2.6% | - |

Table 4: Comparison of VCFLOW with GLFA (Li et al., 2024) and NEURONS (Wang et al., 2025) across subjects. GLFA adopts subject-adaptive pretraining by using fMRI data from all subjects, while NEURONS is trained and evaluated on the same subject.

delivers superior performance in both semantic fidelity and dynamic coherence. In comparison with the subject-specific model NEURONS, which exhibits high visual fidelity and strong temporal consistency, VCFLOW achieves similarly high-quality reconstructions while maintaining robust motion dynamics. Notably, in scenarios involving common semantic structures, VCFLOW even surpasses NEURONS by producing smoother motion trajectories and more semantically accurate content, despite being trained in a subject-agnostic manner.

## C.3 FAILURE CASES

Since our evaluation is conducted in a cross-subject subject-agnostic setting and the amount of training data is limited, a number of failure cases are observed. As illustrated in Fig. 8, the major failure cases can be broadly categorized into two types. The first type arises when the primary objects in the stimulus videos belong to very rare categories, making it difficult for the model to effectively learn their semantics, as exemplified by the first two cases. The second type occurs when the semantics in the stimulus videos are overly complex and intertwined, which makes it challenging to recover the most salient semantic components, as shown in the latter two cases.

## C.4 ADDITIONAL ABLATION RESULTS

To strengthen the persuasiveness of our SARA and HED designs, we conducted ablation studies on both components, as shown in Table 5 and Table 6.

| HCAM | SARA | | | HED | Frame-based | | | | Video-based | | |
|------|------|--|--|-----|-------------|--|--|--|-------------|--|--|
| | $\mathcal{L}_{align}$ | $\mathcal{L}_{subj}$ | $\mathcal{L}_{generic}$ | | 50-way↑ | 2-way↑ | SSIM↑ | PSNR↑ | 50-way↑ | 2-way↑ | CLIP-pcc↑ |
| ✓ | ✓ | | | ✓ | 7.52% | 68.2% | **0.392** | 9.090 | 9.67% | 77.3% | 0.903 |
| ✓ | ✓ | ✓ | | ✓ | 10.7% | 75.0% | 0.368 | 9.983 | 13.9% | 81.7% | 0.924 |
| ✓ | ✓ | ✓ | ✓ | ✓ | **14.2%** | **78.6%** | 0.389 | **10.469** | **18.9%** | **84.8%** | **0.944** |

Table 5: Ablations on the components of SARA, and all results are from subject 1.

## C.5 VISUALIZATION COMPARISON

We conducted a comparative visualization of the preprocessed data and the semantic tokens, as shown in Fig. 9.

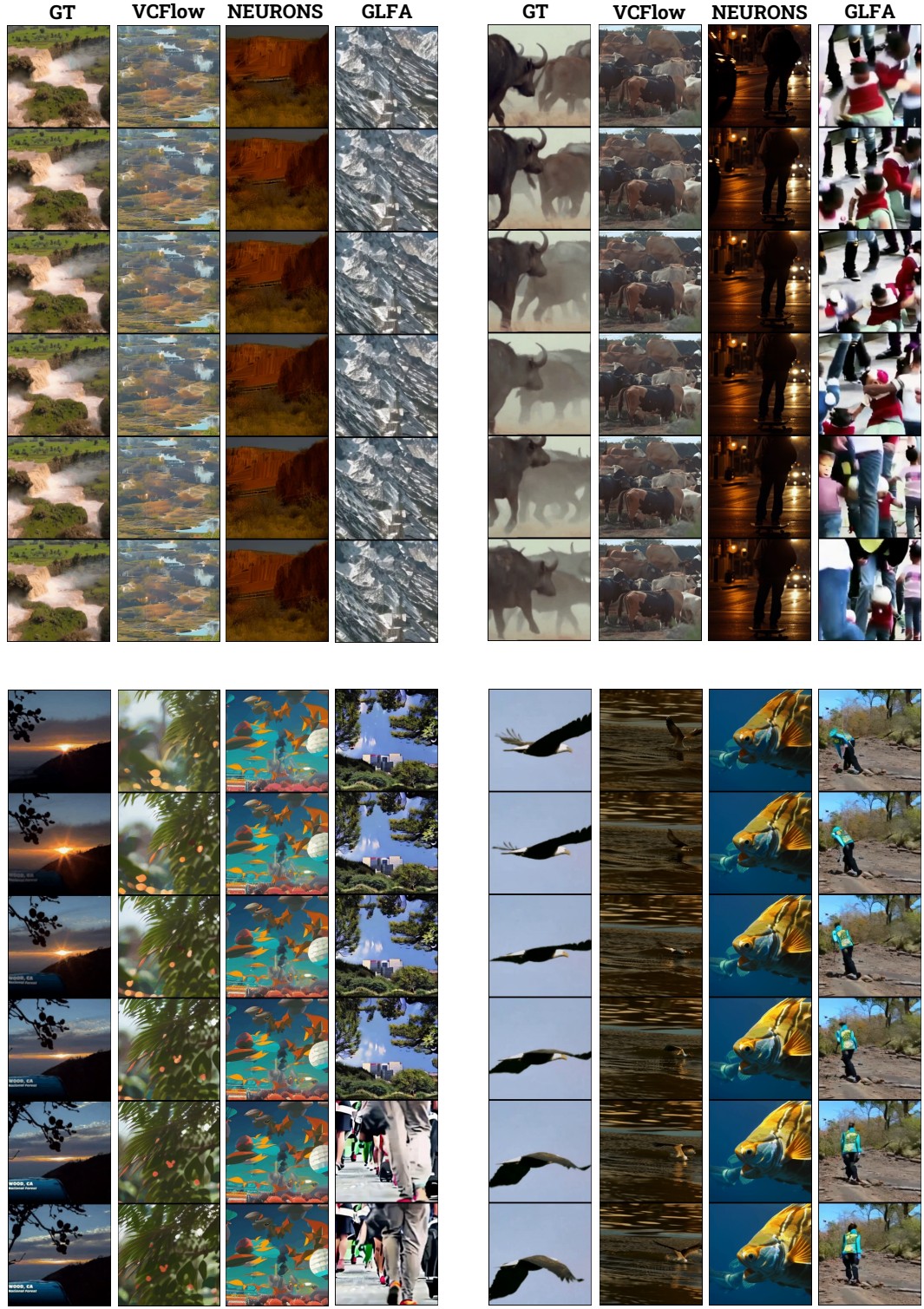

Figure 7: Qualitative comparison results among VCFLOW, GLFA and NEURONS.

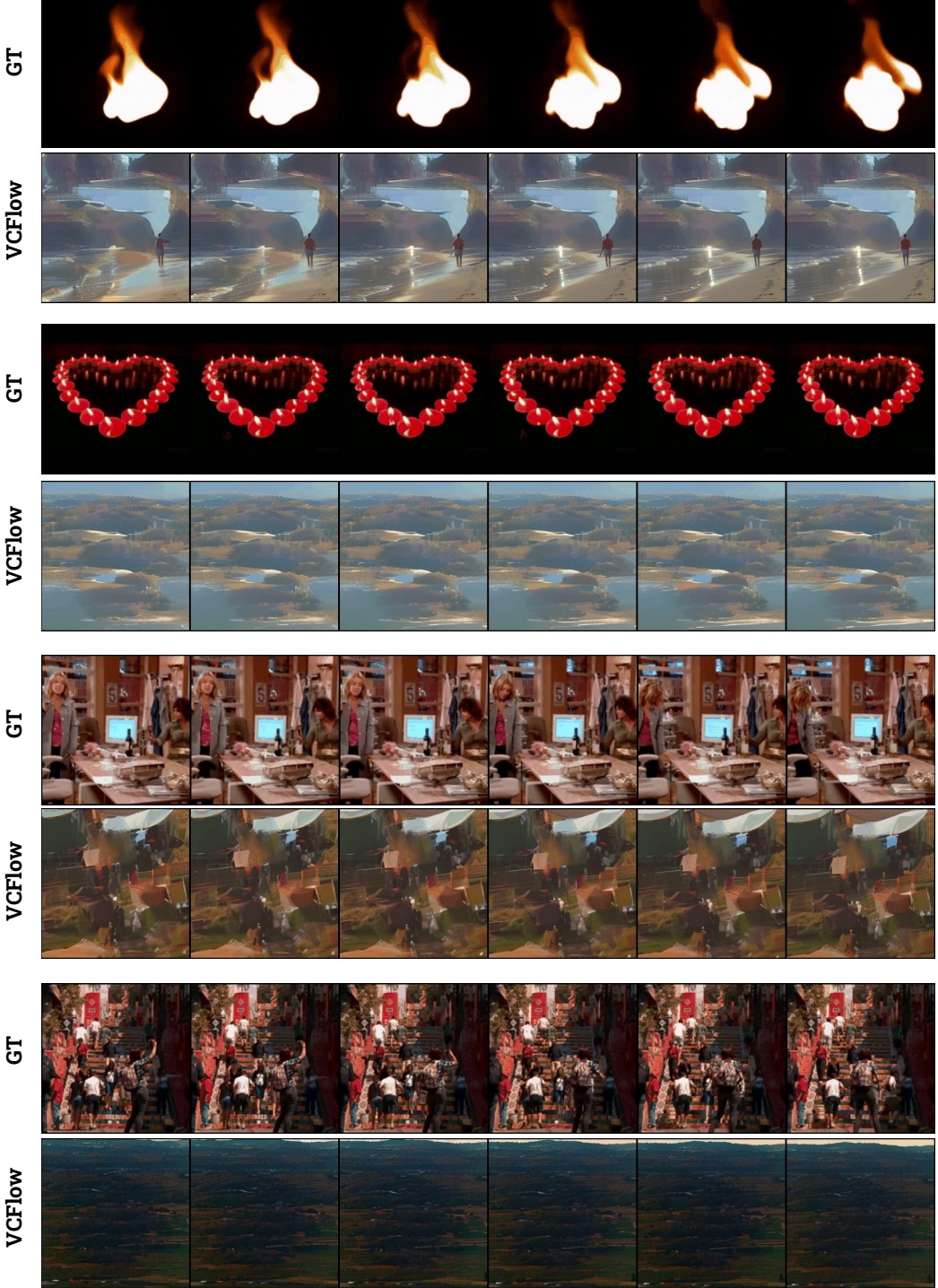

Figure 8: Representative failure cases under the cross-subject subject-agnostic setting, including (1) rare object categories (first two examples) and (2) overly complex and intertwined semantics (last two examples).

## D    USE OF LLMS

Large Language Models (LLMs) were employed solely for spelling and writing refinement in the preparation of this paper. They were not involved in research ideation, methodological design, experimental execution, or substantive content generation.

| HCAM | SARA | HED | | | | | Frame-based | | | | Video-based | | |
|---|---|---|---|---|---|---|---|---|---|---|---|---|---|
| | | $\mathcal{L}_{caption}$ | $\mathcal{L}_{cls}$ | $\mathcal{L}_{seg}$ | $\mathcal{L}_{motion}$ | PL | 50-way↑ | 2-way↑ | SSIM↑ | PSNR↑ | 50-way↑ | 2-way↑ | CLIP-pcc↑ |
| ✓ | ✓ | ✓ | | | | | 10.0% | 72.6% | 0.356 | 9.370 | 12.8% | 81.4% | 0.907 |
| ✓ | ✓ | ✓ | ✓ | | | | 8.2% | 71.1% | 0.360 | 10.197 | 14.6% | 82.3% | **0.970** |
| ✓ | ✓ | ✓ | ✓ | ✓ | | | 13.2% | **78.9%** | **0.408** | 10.737 | 16.4% | 84.1% | 0.942 |
| ✓ | ✓ | ✓ | ✓ | ✓ | ✓ | | 11.3% | 76.0% | 0.383 | 10.123 | 15.2% | 82.4% | 0.926 |
| ✓ | ✓ | ✓ | ✓ | ✓ | ✓ | ✓ | **14.2%** | 78.6% | 0.389 | **10.469** | **18.9%** | **84.8%** | 0.944 |

Table 6: Ablations on the components of HED, and all results are from subject 1.

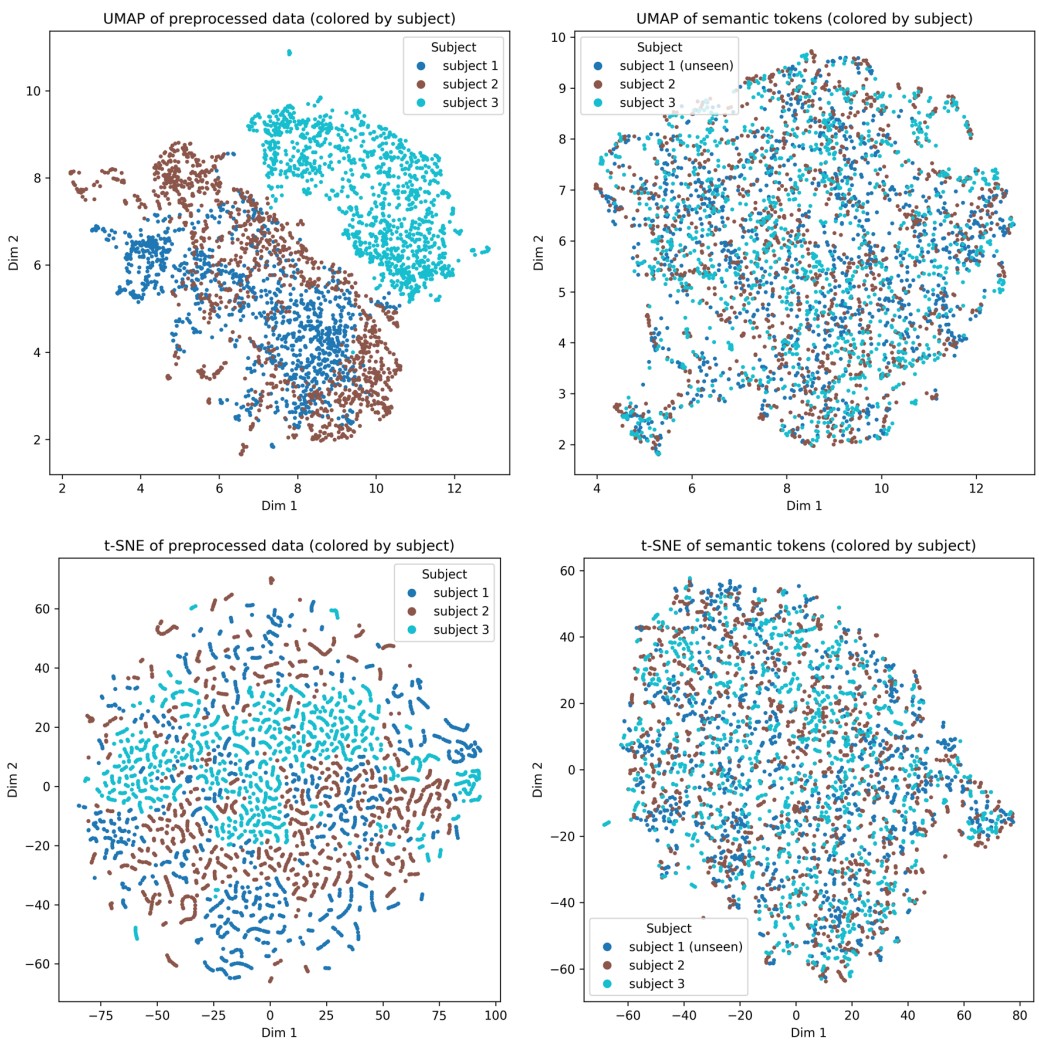

Figure 9: Visualization of the preprocessed data and the semantic tokens.

