# OpenReview forum: "A Cognitive Process-Inspired Architecture for Subject-Agnostic Brain Visual Decoding"
_ICLR.cc/2026/Conference — ICLR 2026 Poster_

### Official Review · Reviewer_ebCw · 2025-10-26

**Soundness:** 3
**Presentation:** 3
**Contribution:** 2
**Rating:** 6
**Confidence:** 3

**Summary:**

This paper proposes VCFLOW, a cognitive process–inspired architecture for cross-subject fmri-to-video decoding. By disentangling early visual, ventral, and dorsal pathways and introducing a SARA module, the method can effectively generalizes to unseen subjects without any finetuning.

**Strengths:**

1. This paper conducts comprehensive ablation studies, and the overall framework design is well visualized.
2. Compared with NEURONS, the proposed SARA module and fine-grained hierarchical alignment mechanism achieve good performance improvements in the cross-subject setting.

**Weaknesses:**

1. The dataset used only contains three subjects, which weakens the persuasiveness of the cross-subject evaluation. Future work could validate on datasets with larger subject pools, such as CineBrain[1].
2. The study only reports results in the cross-subject setting. Given that the overall performance on the fMRI-to-video task is still relatively limited, including single-subject results would better show whether the proposed architecture truly enhances decoding quality rather than merely improving transfer robustness.
3. The claimed 10s inference time appears only in the introduction and lack specific  configuration. The authors should clearly state the experimental conditions and provide comparative results with other methods.
4. Section 3.3 introduces multiple auxiliary tasks, but the paper lacks detailed explanations of how these are constructed and how multi-feature fusion is handled during inference.
5. In terms of neuroscientific interpretation, VCFLOW relies heavily on strong anatomical priors (early/ventral/dorsal separation). Thus, the model validates existing neuroscientific hypotheses rather than discovering such structure in a data-driven way, which limits novelty compared to NEURONS.

[1] CineBrain: A Large-Scale Multi-Modal Brain Dataset During Naturalistic Audiovisual Narrative Processing

**Questions:**

1. How exactly are the video processed and aligned with fMRI signals? More details should be clarified.
2. In Figure 3, what exactly is the “Brain Model” shown in the framework? Additionally, what are the input dimension of the early, ventral, and dorsal features, and what is the dimension of the shared projection space? It would be helpful if these details could be included in the appendix for clarity and reproducibility.

**Details Of Ethics Concerns:**

None.

---

> ### Author Response · Authors · 2025-11-25
> **Response to Reviewer ebCw (1/3)**
>
> # Overall Reply
>
> We sincerely appreciate your positive remarks, including *“comprehensive ablation studies”* and *“achieve good performance improvements,”* which are highly encouraging to us. We will continue refining our work in light of your suggestions. In the following responses, we will clarify each of the points you raised.
>
> # Point-by-point Responses
>
> **W1: about numbers of the subjects**
>
> Thank you for the suggestion. We will conduct experiments on the CineBrain dataset in the future.
>
> **W2: about overall performance on the fMRI-to-video task**
>
> Table 1: Overall performance (the model is trained on subjects 2 and 3)
>
> |  | Frame 50-way | Frame 2-way | SSIM | PSNR | Video 50-way | Video 2-way | CLIP-pcc |
> | --- | --- | --- | --- | --- | --- | --- | --- |
> | Subj 1 | 14.2% | 78.6% | 0.389 | 10.469 | 18.9% | 84.8% | 0.944 |
> | Subj 2 | 14.8% | 78.3% | 0.388 | 10.329 | 18.7% | 84.6% |  0.941 |
> | Subj 3 | 14.2% | 78.1% | 0.383 | 10.326 | 17.9% | 84.7% | 0.944 |
>
> We further evaluated our model on *seen* subjects, as shown in Table 1. The model achieves similar performance on both seen and unseen subjects. This result indicates that the model does not rely on learning subject-specific patterns. Instead, it successfully captures **universal semantic representations** shared across subjects, which explains its strong cross-subject generalizability.
>
> **W3: about details in inference phase**
>
> Thank you for the reminder.
>
> At inference time, our pipeline leverages a pre-trained T2V diffusion model [1], in a setup similar to NEURONS [2]. We condition the model jointly on a control image, a blurry video, and a text description. These three inputs are assembled from the outputs of the decoupled tasks: a control image is reconstructed from each frame via unCLIP [3], while $\mathcal{D}\_{\text{cls}}$ (multi-label classifier) and $\mathcal{D}\_{\text{caption}}$ (text decoder) provide the predicted concepts and generated captions, respectively. The embedding of the top-1 predicted concept, together with the caption embedding, is used to guide video-mask prediction through $\mathcal{D}\_{\text{vs}}$ (segmentation head) and blurry video reconstruction through $\mathcal{D}\_{\text{vr}}$ (reconstruction head). To further enhance the prominence of the key object, we rescale its binary mask from $\{0,1\}$ to $[0.5,1]$ and multiply it with both the control image and the blurry video before feeding them into the diffusion model.
>
> Because inference for unseen subjects does not require any subject-specific finetuning, their data can be directly processed by the model. With this setup, the inference time—normalized by the total number of videos—remains below 10 seconds per video. All measurements are obtained on an NVIDIA RTX 4090 GPU.
>
> We have also included this in the Appendix A.6.
>
> From a comparative perspective, existing methods require retraining on each new subject before inference, which incurs substantial computational overhead and long processing times. In our task, subject-specific models typically require **around 12 hours of subject-specific training per unseen subject.** In contrast, our model can directly perform inference on unseen subjects without any additional training, and the inference for each video takes only 10 seconds.

---

> ### Author Response · Authors · 2025-11-25
> **Response to Reviewer ebCw (2/3)**
>
> **W4: about detailed explanations of auxiliary tasks**
>
> Thank you for your question.
>
> We first present the ventral-related tasks, which are mainly designed to capture high-level semantic information along the visual processing stream.
>
> **Concept Recognition.**
> To enhance conceptual understanding, we introduce a concept recognition task by adding a multi-label classifier $\mathcal{D}\_{\text{cls}}(\cdot)$ that predicts the key concepts in each frame from the fMRI-derived visual embeddings. Concretely, we apply a cross-entropy loss between the classifier prediction and the ground-truth (GT) concept list:
> $\mathcal{L}\_{\text{cls}} = \mathcal{L}\_{\text{ce}}(\mathcal{D}\_{\text{cls}}(\bar{e}^v), \mathcal{C}),$
> where $\bar{e}^v$ denotes the mean of ${e}^v$ along the frame axis, and $\mathcal{C}$ is the GT concept list.
>
> **Scene Description.**
> To further model scene-level semantics, we incorporate a scene description task that aims to generate a descriptive caption for each video frame. Specifically, we finetune a pre-trained text decoder $\mathcal{D}\_{\text{caption}}(\cdot)$, which takes the fMRI-derived text embeddings $e^t$ as input and produces the caption. We adopt GPT-2 [4] as the text decoder and train it using prefix language modeling. Given a GT caption token sequence $\mathcal{S} = \{s\_0, s\_1, \dots, s\_{\lvert\mathcal{S}\rvert}\}$ and the corresponding text embedding $e^t$, the decoder $\mathcal{D}\_{\text{caption}}(\cdot)$ is trained to reconstruct $\mathcal{S}$ conditioned on $e^t$ as the prefix. The training objective $\mathcal{L}\_{\text{caption}}$ is defined as:
> $\mathcal{L}\_{\text{caption}} = -\frac{1}{\lvert\mathcal{S}\rvert} \sum^{\lvert\mathcal{S}\rvert}\_{i=1} \log \mathcal{D}\_{\text{caption}}(s\_i \mid s\_{<i}, e^t),$
> where $s\_i$ denotes the $i$-th token in the GT sequence $\mathcal{S}$.
>
> Next, we consider early-vision tasks, which primarily aim to learn coarse object contours and spatial masks.
>
> **Key Object Segmentation.**
> To better capture object-level information, we design a text-driven video decoder based on the VAE video decoder [5]. This decoder takes the video embeddings $e^v$ together with the text embeddings $e^t$ as inputs. For this task, the text embeddings are obtained by encoding the category name of the key object with the CLIP text encoder. A cross-attention module is then applied to activate specific patches in $e^v$ (used as queries $Q$) corresponding to $e^t$ (used as keys $K$ and values $V$):
> ${e}^{\text{seg}} = \mathrm{softmax}\left(\frac{Q K^\top}{\sqrt{d}}\right) \cdot V.$
>
> The activated feature ${e}^{\text{seg}}$ is upsampled to a higher resolution for pixel-level prediction. We then employ a simple segmentation head $\mathcal{D}\_{\text{vs}}(\cdot)$ to generate the binary segmentation masks $y\_{\text{seg}}$ for the key object in the video. The training objective for this task is a binary cross-entropy loss, denoted as $\mathcal{L}\_{\text{seg}}$.
>
> Finally, we introduce the dorsal-related task, which focuses on modeling global motion information.
>
> **Blurry Video Reconstruction.**
> We reuse the same VAE decoder as in the key object segmentation task, but replace the segmentation head $\mathcal{D}\_{\text{vs}}(\cdot)$ with a reconstruction head $\mathcal{D}\_{\text{vr}}(\cdot)$. Given the video embeddings $e^v$, the text-driven video decoder together with $\mathcal{D}\_{\text{vr}}(\cdot)$ generates a blurry video $y^{\text{motion}}\_c$. We then map $y^{\text{motion}}\_c$ into the latent space of the Stable Diffusion VAE to obtain the latent embeddings $y'\_c$. This subtask is trained with a mean absolute error (MAE) loss, defined as:
> $\mathcal{L}\_{\text{motion}} = \frac{1}{F} \sum^{F}\_{i=1} \lvert y^{\text{motion}}\_{c,i} - y'\_{c,i} \rvert.$
>
> We have also included these descriptions in the Appendix A.5. If you would like to examine additional fine-grained parameter settings, you may also consult our code.

---

> ### Author Response · Authors · 2025-11-25
> **Response to Reviewer ebCw (3/3)**
>
> **W5: about the reliance on anatomical priors and its limitation in novelty**
>
> Thank you for your question. However, we respectfully disagree with the claim that “the model validates existing neuroscientific hypotheses rather than discovering such structure in a data-driven way”.
>
> While our method does incorporate anatomical priors, these priors function only as a coarse, conceptual guideline inspired by high-level cognitive processing pathways. They do *not* impose any strict or explicit definition of which specific regions belong to the dorsal, ventral, or early visual streams. Therefore, a key challenge lies in how to operationalize such priors from the data level and how to integrate them into the model architecture—this is far from a simple or rigid anatomical prior, but rather a principled data-driven incorporation of cognitive structure.
>
> As discussed in the Appendix A.3, our experiments include analyses of different region-parcellation strategies, and the results demonstrate that our model **effectively handles the inherent ambiguity** of these anatomical divisions in both statistical and empirical terms. In this sense, although the prior provides a rough conceptual structure, the final modeling behavior is largely data-driven.
>
> **Scheme A**
>
> - **Early vision:** V1, V2, V3, V4
> - **Dorsal stream:** V3A, V3B, V6, V6A, V7, IPS1, LO1, LO2, LO3, FST, MT, MST, V3CD, V4t, PH, IP0
> - **Ventral stream:** FFC, PIT, V8, VMV1, VMV2, VMV3, VVC, PHA1, PHA2, PHA3, TE2p
>
> **Scheme B**
>
> - **Early vision:** V1, V2, V3, V4
> - **Dorsal stream:** V3A, V3B, V6, V6A, V7, IPS1, FST, MT, MST, V3CD, V4t, IP0
> - **Ventral stream:** FFC, PIT, V8, VMV1, VMV2, VMV3, VVC, PHA1, PHA2, PHA3, TE2p, LO1, LO2, LO3, PH
>
> **ROI Scheme Comparison (Subject 1)**
>
> | Scheme | **50-way ↑** | **2-way ↑** | **SSIM ↑** | **PSNR ↑** | **50-way ↑** | **2-way ↑** | **CLIP-pcc ↑** |
> | --- | --- | --- | --- | --- | --- | --- | --- |
> | **Scheme A** | **14.2%** | **78.6%** | **0.389** | **10.469** | **18.9%** | **84.8%** | **0.944** |
> | Scheme B | 12.4% | 71.9% | 0.353 | 9.366 | 9.8% | 80.6% | 0.913 |
>
> **Q1: about the details of how the video processed and aligned with fMRI signals**
>
> Thank you for the reminder. Please refer to w4.
>
> **Q2: about details in model configurations**
>
> Thank you for the reminder.
>
> As shown in Fig.3, our Brain Model extracts four types of representations from the input fMRI signals: a full-brain representation $ \mathbf{E}\_{\text{brain}} $, early visual features $ \mathbf{E}\_{\text{early}} $, ventral-stream features $ \mathbf{E}\_{\text{ventral}} $, and dorsal-stream features $ \mathbf{E}\_{\text{dorsal}} $. Here, the Brain Model refers to the module responsible for deriving these fMRI-based features, consisting of a ViT backbone together with several linear projection heads.
>
> The linear heads operate directly on the flattened voxel signals to obtain the three cognitively grounded feature sets—$ \mathbf{E}\_{\text{early}} $, $ \mathbf{E}\_{\text{ventral}} $, and $ \mathbf{E}\_{\text{dorsal}} $—each with shape $ \mathbb{R}^{B \times S' \times D} $. For the global full-brain representation $ \mathbf{E}\_{\text{brain}} $, we adopt a ViT-based fMRI encoder (fMRI-PTE [6], pretrained on the UK Biobank dataset [7]), which produces features of shape $ \mathbb{R}^{B \times S \times D} $.
>
> We have also included this in Appendix A.7.  For further clarification regarding the exact parameter settings, you may additionally refer to our code.
>
> [1] Guo, Yuwei, et al. "Animatediff: Animate your personalized text-to-image diffusion models without specific tuning." arXiv preprint arXiv:2307.04725 (2023).
>
> [2] Wang, Haonan, et al. "Neurons: Emulating the Human Visual Cortex Improves Fidelity and Interpretability in fMRI-to-Video Reconstruction." arXiv preprint arXiv:2503.11167 (2025).
>
> [3] Ramesh, Aditya, et al. "Hierarchical text-conditional image generation with clip latents." arXiv preprint arXiv:2204.06125 1.2 (2022): 3.
>
> [4] Radford, Alec, et al. "Language models are unsupervised multitask learners." OpenAI blog 1.8 (2019): 9.
>
> [5] Von Platen, Patrick, et al. "Diffusers: State-of-the-art diffusion models." 2022,
>
> [6] Qian, Xuelin, et al. "fmri-pte: A large-scale fmri pretrained transformer encoder for multi-subject brain activity decoding." arXiv preprint arXiv:2311.00342 (2023).
>
> [7] Miller, Karla L., et al. "Multimodal population brain imaging in the UK Biobank prospective epidemiological study." Nature neuroscience 19.11 (2016): 1523-1536.

---

> ### Author Response · Authors · 2025-11-28
> **Dear Reviewer ebCw**
>
> Thank you again for your constructive feedback. With the discussion period drawing to a close, we would be grateful to know whether our rebuttal has sufficiently addressed your concerns. We are happy to respond promptly to any additional questions you may have.

---

### Official Review · Reviewer_y215 · 2025-10-30

**Soundness:** 3
**Presentation:** 3
**Contribution:** 3
**Rating:** 6
**Confidence:** 4

**Summary:**

This paper proposes the Visual Cortex Flow Architecture (VCFLOW), a hierarchical brain decoding framework capable of learning multi-dimensional representations from different brain regions (early visual cortex, ventral, and dorsal streams). It also introduces a feature-level contrastive learning strategy to help the model generalize to unseen subjects. The article achieves this by sacrificing a certain degree of accuracy, enabling the model to perform well on unseen test subjects.

**Strengths:**

1. The motivation of the article is very good. I completely agree that the brain decoding model should further focus on decoding for multiple individuals, especially for unseen subjects. Because we cannot expect users to undergo prolonged collection processes in practical use. Generalization among the subjects is a problem that must be addressed.

2. The article is well-written and clearly expressed. The experiment was also conducted thoroughly.

3. The model not only designs multiple different modules to extract features at different levels for alignment, but also adopts multiple tasks to conduct a more comprehensive alignment.

**Weaknesses:**

1. I agree that generalization to unseen subjects is very important. However, I do not fully agree that the reliability of the method can be demonstrated in the video reconstruction task. As is well known, current brain video reconstruction largely depends on the pre-trained Diffusion model to achieve its effects. Then, we calculate the indicators based on the generated videos. I believe the results are heavily influenced by the generation model rather than actual brain decoding. (In other words, brain decoding might only produce very high-dimensional and not detailed semantic information, which the diffusion model then uses to generate the video. In fact, the results shown in Figure 5 also reflect this point. Current brain decoding is likely to have provided only very vague information. We evaluate everything based on the generated video, and there are too many factors in between that can influence the outcome.)

2. Based on the article's indicators, the improvement is minimal. In the 50-way scenario, it only increased by 2% to 4% compared to other pre-training methods, resulting in just 2 to 4 more images correctly identified out of 100 samples. Additionally, the pixel-level metrics are very low, so they offer limited comparative value. Even a completely black image can reach a PSNR of around 10.

3. The study had too few subjects. With only three participants involved, it is very difficult to rule out coincidental factors and establish a clear pattern. The works Clip-Mused [1] and TGBD [2] focus on retrieving image/video frames based on brain signals for multiple subjects. Particularly, TGBD is also designed for unseen subjects, utilizing the HCP[3] dataset (with over 170 participants). The main conclusion is that as the number of participants increased, the generalization ability on unseen participants significantly improved. I think TGBD could provide inspiration for this work and should be included by related works, as one of the few studies that focused on the unseen subjects. Maybe the author's method is limited by the number of participants. When the number of participants is sufficient, a much greater performance gap might be achieved.


[1] CLIP-MUSED: CLIP-guided multi-subject visual neural information semantic decoding, ICLR2024

[2] Toward Generalizing Visual Brain Decoding to Unseen Subjects, ICLR2025

[3] The WU-Minn Human Connectome Project: An Overview. Neuroimage, 2013.

**Questions:**

Please see Weaknesses section.

---

> ### Author Response · Authors · 2025-11-25
> **Response to Reviewer y215**
>
> # Overall Reply
>
> Thank you very much for your positive comments, such as *“The motivation of the article is very good,”* *“The experiments were conducted thoroughly,”* and *“conduct a more comprehensive alignment.”* We are especially grateful that your recognition of our motivation aligns with the views of other reviewers. Your encouragement is greatly appreciated, and we will address each of the concerns you raised in detail.
>
> # Point-by-point Responses
>
> **W1: about whether the reliability of the method can be demonstrated in the video reconstruction task**
>
> Thank you for your question. However, we respectfully disagree with the claim that *“the results are heavily influenced by the generation model rather than actual brain decoding.”*
>
> - **Difference between reconstruction task and other tasks:** We believe that **video reconstruction** fundamentally differs from **brain-decoding classification or retrieval tasks**. Compared with classification or retrieval—which only require coarse semantic information to achieve discriminative separability—**video reconstruction demands much higher semantic completeness and low-dimensional structural guidance**, neither of which can be filled in solely by the diffusion model.
> - **The high semantic completeness in our results:** As shown in our figures, the results go beyond simple category distinction. They exhibit **coherent semantic reconstruction** that must have been decoded from brain activity.
>     - In Figure 1, it captures the concept of *“two people walking along a seaside beach.”*
>     - In Figure 2, it captures the concept of *“an airplane flying in a blue sky with clouds.”*
>     - In Figure 3, it captures the concept of *“A couple sweetly leaning against each other”*
>     - In Figure 4, it captures the concept of *“A person in a swimming pool”*
>
>     Such **semantically complete and structurally consistent** reconstructions cannot be produced by the diffusion prior alone; they require **precise semantic information decoded from the brain**.
>
> - **Low-level information can also be decoded:** Regarding fine-grained details, video decoding is inherently challenging, and the difficulty is further compounded by the requirement to **generalize to unseen subjects**. As a result, some visual details may not be perfectly preserved. Nevertheless, the **color composition and global contour structure** observed in the reconstructions indicate that the model relies on more than just vague signals and a diffusion prior—these aspects reflect meaningful information extracted directly from neural data.
>
> **W2: about the improvement in metrics**
>
> Thank you for the suggestion. However, we believe there may be some misunderstanding regarding the interpretation of these metrics.
>
> - **PSNR remains a meaningful comparative metric despite the trivial baseline of a black image：** While it is true that a completely black image may obtain a PSNR around 10, this does not imply that pixel-level metrics lack comparative value. PSNR is a *relative* metric that captures the magnitude of reconstruction error, and even small differences in PSNR correspond to substantial differences in pixel-wise deviations. In our setting, the reconstructed videos consistently achieve PSNR values significantly higher than the trivial baseline, indicating that the model preserves meaningful low-frequency structural information.
> - **The 50-way top-1 evaluation is inherently a highly challenging task that requires fine-grained semantic understanding:** We believe there is also a misunderstanding regarding the difficulty of the 50-way top-1 evaluation. A 50-way top-1 task is inherently challenging and requires highly refined semantic representations. If the selection were random, the expected accuracy would be only **2%**. Therefore, achieving a 30–40% improvement over this baseline represents a substantial gain, indicating that the model indeed captures meaningful semantic information.
>
> **W3: about numbers of the subjects**
>
> Thank you for your question. Indeed, due to the limitations of the fMRI2Video dataset, the number of available subjects cannot be very large. TGBD is certainly an insightful and relevant work, and we will include it in the related work discussion—thank you for pointing this out. From the conclusions of TBGD, We believe that if the model can achieve the current performance with only a few subjects, its performance would likely improve even further when trained with more subjects.

---

> ### Author Response · Authors · 2025-11-28
> **Dear Reviewer y215**
>
> Thank you again for your constructive feedback. With the discussion period drawing to a close, we would be grateful to know whether our rebuttal has sufficiently addressed your concerns. We are happy to respond promptly to any additional questions you may have.

---

### Official Review · Reviewer_HxmM · 2025-10-31

**Soundness:** 2
**Presentation:** 2
**Contribution:** 3
**Rating:** 4
**Confidence:** 3

**Summary:**

This paper introduces Visual Cortex Flow Architecture (VCFLOW), which is a subject-agnostic framework for reconstructing videos from fMRI.  One key idea of the pipeline is to extract hierarchical features from brain regions which are the early visual areas, ventral and dorsal stream areas and  align them with different types of visual representations to capture complementary information such as low-level, semantic and motion cues. Besides, the authors design a subject-agnostic model by proposing a module called SARA that projects subject-specific information into a shared space.

**Strengths:**

- The paper is well-written and mostly easy to follow (i.e up to the complexity of the presented method)
- The motivation for subject-agnostic model is clearly explained and detailed.
- Most fMRI-to-video decoding methods are subject-specific which is a significant bottleneck for clinical applications while this method addresses this fundamental challenge, making it a first important step towards better practicality.

**Weaknesses:**

- The method seems quite complex with many components while the ablation seems quite incomplete and doesn’t ensure all the elements are crucial to the final performance.
- Missing information on how they generate the final videos in less than 10 seconds (claim in abstract). Could they be more specific into which generative model they use and exactly which brain-predicted embeddings they give to it (as only stable diffusion is mentioned in figure 3 but without further details, maybe explained in fig 4 but didn’t really understand... )?
- Difficult to draw parallels between the letters/concepts introduced in method section 3 and drawing in figure 3. Could you represent on fig 3 at which stage E_early, E_ventral, E_dorsal and F_brain, F_ventral, T_sem, T_subj are present in the figure for instance ?
- The authors claim ‘A key innovation of our approach lies in the utilization of CLIP embeddings from multiple layers to achieve fine-grained semantic alignment with fMRI signals’ but it has already been done in previous brain-to-image papers.

**Questions:**

- Could the authors explain more clearly evaluation metrics ? N-way top k is cited for frame-level metrics but it seems to also be used for semantic accuracy in video metrics
- Could you add an ablation on the different SARA/HED losses displayed in line 267 and line 298, on the effect of adjusting the loss coefficients in line 299
- Could you give more details on what is the image pretraining phase ? To which representations do they align brain features ? ( as i guess there is no motion representations for instance)
- Would be nice to compare to other baselines such as the recent work ANIMATE YOUR THOUGHTS: RECONSTRUCTION OF  DYNAMIC NATURAL VISION FROM HUMAN BRAIN ACTIVITY (ICLR25)

---

> ### Author Response · Authors · 2025-11-25
> **Response to Reviewer HxmM (1/3)**
>
> # Overall Reply
>
> We sincerely appreciate your thoughtful feedback, including comments such as *“well-written and mostly easy to follow,”* *“the motivation for the subject-agnostic model is clearly explained and detailed,”* and *“this method addresses a fundamental challenge.”* Your positive evaluation is highly encouraging to us. We will also clarify the specific points you raised in your review to ensure full transparency and understanding.
>
> # Point-by-point Responses
>
> **W1: about  components in the framework**
>
> Thank you for your question. Although our framework may appear complex, each component is **purposeful and effective**. Every task is deliberately designed to capture a **complementary dimension of information**, and together they form a unified and coherent architecture.
>
> Below, we present the additional ablation results.
>
> Table 1: Ablations on the components of SARA (Subject 1)
>
> | SARA L_align | SARA L_subj | SARA L_generic | Frame 50-way ↑ | Frame 2-way ↑ | Frame SSIM ↑ | Frame PSNR ↑ | Video 50-way ↑ | Video 2-way ↑ | Video CLIP-pcc ↑ |
> | --- | --- | --- | --- | --- | --- | --- | --- | --- | --- |
> | ✓ |  |  | 7.52% | 68.2% | **0.392** | 9.090 | 9.67% | 77.3% | 0.903 |
> | ✓ | ✓ |  | 10.7% | 75.0% | 0.368 | 9.983 | 13.9% | 81.7% | 0.924 |
> | ✓ | ✓ | ✓ | **14.2%** | **78.6%** | 0.389 | **10.469** | **18.9%** | **84.8%** | **0.944** |
>
> Table 2: Ablations on the components of HED (Subject 1, PL refers to “progressive learning”)
>
> | HED L_caption | HED L_cls | HED L_seg | HED L_motion | HED PL | Frame 50-way ↑ | Frame 2-way ↑ | Frame SSIM ↑ | Frame PSNR ↑ | Video 50-way ↑ | Video 2-way ↑ | Video CLIP-pcc ↑ |
> | --- | --- | --- | --- | --- | --- | --- | --- | --- | --- | --- | --- |
> | ✓ |  |  |  |  | 10.0% | 72.6% | 0.356 | 9.370 | 12.8% | 81.4% | 0.907 |
> | ✓ | ✓ |  |  |  | 8.2% | 71.1% | 0.360 | 10.197 | 14.6% | 82.3% | **0.970** |
> | ✓ | ✓ | ✓ |  |  | 13.2% | **78.9%** | **0.408** | 10.737 | 16.4% | 84.1% | 0.942 |
> | ✓ | ✓ | ✓ | ✓ |  | 11.3% | 76.0% | 0.383 | 10.123 | 15.2% | 82.4% | 0.926 |
> | ✓ | ✓ | ✓ | ✓ | ✓ | **14.2%** | 78.6% | 0.389 | **10.469** | **18.9%** | **84.8%** | 0.944 |
>
> We have also included these results in Appendix C.4.
>
> These ablation results clearly demonstrate that each module contributes critically to **semantic modeling**, confirming that every component in our framework is both **purposeful and effective**.
>
> **W2: about video generation in the inference stage**
>
> Thank you for your reminder.
>
> Following the functional roles of each visual pathway, the ventral stream feature is used for the semantic modules, including the multi-label classifier $\mathcal{D}\_{\text{cls}}$ and the text decoder $\mathcal{D}\_{\text{caption}}$. The early visual feature supervises the low-level segmentation head $\mathcal{D}\_{\text{vs}}$, while the dorsal stream feature guides the reconstruction head $\mathcal{D}\_{\text{vr}}$.
>
> At inference time, our pipeline leverages a pre-trained T2V diffusion model [1], in a setup similar to NEURONS [2]. We condition the model jointly on a control image, a blurry video, and a text description. These three inputs are assembled from the outputs of the decoupled tasks: a control image is reconstructed from each frame via unCLIP [3], while $\mathcal{D}\_{\text{cls}}$ (multi-label classifier) and $\mathcal{D}\_{\text{caption}}$ (text decoder) provide the predicted concepts and generated captions, respectively. The embedding of the top-1 predicted concept, together with the caption embedding, is used to guide video-mask prediction through $\mathcal{D}\_{\text{vs}}$ (segmentation head) and blurry video reconstruction through $\mathcal{D}\_{\text{vr}}$ (reconstruction head). To further enhance the prominence of the key object, we rescale its binary mask from $\{0,1\}$ to $[0.5,1]$ and multiply it with both the control image and the blurry video before feeding them into the diffusion model.
>
> Because inference for unseen subjects does not require any subject-specific finetuning, their data can be directly processed by the model. With this setup, the inference time—normalized by the total number of videos—remains below 10 seconds per video. All measurements are obtained on an NVIDIA RTX 4090 GPU.
>
> We have also included it in Appendix A.6.

---

> ### Author Response · Authors · 2025-11-25
> **Response to Reviewer HxmM (2/3)**
>
> **W3: about the letters/concepts introduced in method section 3 and drawing in figure 3.**
>
> Thank you for your question, we apologize for the confusion caused. We have incorporated these symbols directly into the figures to make the overall process clearer and easier to follow.
>
> **W4: about previous work of utilization of CLIP embeddings from multiple layers**
>
> We thank the reviewer for the careful evaluation. We agree that the original sentence may be confusing, and we will revise it as follows:
>
> “*A key innovation of our approach is that, by aligning the dual-stream cognitive hierarchy of the brain with the hierarchical structure of CLIP, we construct a universal cognitive structure that enhances cross-subject generalization.*”
>
> Existing fMRI-to-image reconstruction frameworks commonly follow a similar design philosophy: aligning fMRI-derived embeddings to different semantic levels of a CLIP model. For instance, **MindEye2** [4] and **Mindtuner** [5] maps fMRI features directly into CLIP space to obtain high-level semantics, while relying on a blurry-image reconstruction module to recover low-level visual details. **NeuroPictor** [6] instead aligns fMRI to CLIP text embeddings to capture conceptual representations, and learns low-level features by finetuning Stable Diffusion with fMRI embeddings as additional conditioning.
>
> **However, despite these attempts to represent multiple semantic levels, their designs fundamentally differ from ours and are not suitable for subject-agnostic cross-subject generalization.**
>
> **(1) They do not explicitly use *multi-level* CLIP features for hierarchical alignment.**
>
> Most existing models map fMRI into a *single* global CLIP space (text or image), overlooking the fact that CLIP’s internal layers naturally form a hierarchical semantic structure that resembles the gradual abstraction in the human visual cortex. This mismatch prevents them from fully leveraging cross-level representation consistency between brains and CLIP.
>
> **(2) They extract all levels of semantics from a *single, holistic* brain embedding.**
>
> Using one global embedding to express both high-level semantics and low-level perceptual information forces the model to mix subject-specific idiosyncrasies with universal representations. As a result, these approaches struggle to learn subject-invariant features and thus fail to generalize in the subject-agnostic setting—where no subject-specific retraining is allowed.
>
> **In contrast, our method not only extracts hierarchical semantics from multi-level CLIP features, but also introduces an explicit analogy to the brain’s own cognitive processing streams.**
>
> By grounding the model architecture in a **universal, subject-shared cortical hierarchy**, our framework separates subject-invariant semantic information from subject-specific noise, enabling robust and scalable subject-agnostic generalization—something existing methods were not designed to achieve.

---

> ### Author Response · Authors · 2025-11-25
> **Response to Reviewer HxmM (3/3)**
>
> **Q1: about evaluation metrics**
>
> We apologize for the ambiguity in our original description. The N-way top-K metric is indeed used at both the video level and the image level, but the difference lies in the label sets used for evaluation.
>
> At the frame level, we perform an N-way top-K classification task based on 1,000 ImageNet categories. At the video level, Semantic accuracy is also measured through an N-way top-K action classification task (top 1 in our case) involving 400 classes from the Kinetics-400 dataset, using a VideoMAE-based model as the classifier.
>
> We have revised the evaluation section to make the distinction clearer and ensure that the overall metric description is more precise and understandable.
>
> **Q2: about additional ablation results**
>
> Thank you for your question, please refer to w1.
>
> **Q3: about the pretraining phase**
>
> Thank you for your reminder.
>
> Due to the lack of dynamic information in image datasets, the primary goal of our pretraining stage is to train the backbone to acquire an initial understanding of semantics and to develop the ability to separate subject tokens. Therefore, we adopt the SARA training objective, as shown in Equation 7.
>
> We have also incorporated this explanation into Section 4.1 of the main text. If you would like to examine additional fine-grained parameter settings, you may also consult our code.
>
> **Q4: about other baselines**
>
> Thank you for your reminder. Since our work primarily focuses on generalization to unseen subjects, certain models that are difficult to adapt to our evaluation setting were not included in our comparisons. Nonetheless, this is certainly a valuable subject-specific related work, and we will include it in the related work discussion. Thank you for bringing this to our attention.
>
> [1] Guo, Yuwei, et al. "Animatediff: Animate your personalized text-to-image diffusion models without specific tuning." arXiv preprint arXiv:2307.04725 (2023).
>
> [2] Wang, Haonan, et al. "Neurons: Emulating the Human Visual Cortex Improves Fidelity and Interpretability in fMRI-to-Video Reconstruction." arXiv preprint arXiv:2503.11167 (2025).
>
> [3] Ramesh, Aditya, et al. "Hierarchical text-conditional image generation with clip latents." arXiv preprint arXiv:2204.06125 1.2 (2022): 3.
>
> [4] Scotti, Paul S., et al. "Mindeye2: Shared-subject models enable fmri-to-image with 1 hour of data." arXiv preprint arXiv:2403.11207 (2024).
>
> [5] Gong, Zixuan, et al. "Mindtuner: Cross-subject visual decoding with visual fingerprint and semantic correction." Proceedings of the AAAI Conference on Artificial Intelligence. Vol. 39. No. 13. 2025.
>
> [6] Huo, Jingyang, et al. "Neuropictor: Refining fmri-to-image reconstruction via multi-individual pretraining and multi-level modulation." European Conference on Computer Vision. Cham: Springer Nature Switzerland, 2024.

---

> ### Author Response · Authors · 2025-11-28
> **Dear Reviewer HxmM**
>
> Thank you again for your constructive feedback. With the discussion period drawing to a close, we would be grateful to know whether our rebuttal has sufficiently addressed your concerns. We are happy to respond promptly to any additional questions you may have.

---

### Official Review · Reviewer_9GCm · 2025-10-31

**Soundness:** 2
**Presentation:** 1
**Contribution:** 2
**Rating:** 4
**Confidence:** 4

**Summary:**

I have carefully read this manuscript. The research area of this paper is fMRI-to-video reconstruction, with a focus on the challenge of cross-subject transfer (generalization to new/unseen subjects). This is an important problem, as the authors describe in the introduction: in practical applications, it is impossible to collect large amounts of data from a single subject for training decoding models. Although the authors emphasize the cross-subject issue, only about one-third of the proposed methods (Section 3.2) are actually related to this problem, with the majority focusing on designing fMRI model architectures that better align with human visual mechanisms and on improving representation learning methods.

Furthermore, I believe the paper lacks certain methodological and evaluation details regarding cross-subject transfer; the authors do not explain how the model adapts when new subjects are introduced. Putting these issues aside, I think the contribution of the methods proposed in this paper to cross-subject visual decoding is still limited.

**Strengths:**

+ This paper provides a good overview of the work on fMRI-to-video reconstruction.

+ This paper focuses on an important issue.

**Weaknesses:**

1. The abstract and the first two paragraphs of the introduction strongly emphasize the issue of transfer/generalization to unseen subjects. However, in the third paragraph of the introduction (lines 73–94), the methods described by the authors have no relation to this issue. This creates a jarring shift in the writing. Similarly, after reading the Introduction, I think the authors should focus on how to achieve transfer to unseen subjects. Yet, in the methods section, only one design is actually related to this issue. Therefore, I believe the introduction of the manuscript requires substantial revision and should not solely emphasize the transfer to unseen subjects.

2. I am fairly familiar with the related work. Essentially, the token extension method proposed by the authors in Section 3.2 assigns subject-specific tokens to each participant. This approach has actually appeared previously in fMRI-to-Image studies [1]. However, it should be acknowledged that in the fMRI-to-Video task, the authors may indeed be the first to apply this method.

3. I believe that some of the methodological and experimental descriptions in this manuscript are not sufficiently clear, and I could not understand how the model adapts when a new subject is introduced; for details, please refer to the Questions section.

4. The citation format in the main text of this manuscript needs to be corrected.

5. According to Figure 5, the video reconstruction results are not very satisfactory.

[1] Zhou et al. CLIP-MUSED: CLIP-guided multi-subject visual neural information semantic decoding. ICLR 2024.

**Questions:**

1. In Equation 2, does 𝐹 refer to the fMRI voxels or the result obtained through the diffusion prior? According to the equation, it seems to be the former, but according to Figure 3, it appears to be the latter.

2. Equation 5 is puzzling—why is the InfoNCE loss computed using fMRI representations from two different subjects within the batch? How is the InfoNCE loss computed with only two samples?

3. What do 𝑧 and 𝑦 represent in Equation 6?

4. The authors state in line 101 that no training using new-subject data is needed. Does this mean the new subject’s test data is fed directly into the model? If so, where does the subject-specific extended token T_{subj} come from?

---

> ### Author Response · Authors · 2025-11-25
> **Response to Reviewer 9GCm (1/3)**
>
> # Overall Reply
>
> We sincerely thank you for your positive assessment, including remarks such as *“a good overview of the work on fMRI-to-video reconstruction”* and *“focuses on an important issue.”* We are particularly grateful for your constructive feedback, which affirms the practical value of our approach. While certain details may indeed require clarification, we will address these points in the following responses to ensure that our methodology and results are clearly and accurately understood.
>
> # Point-by-point Responses
>
> **W1: about the motivation presented in the introduction and how our method is designed to accomplish it**
>
> We thank the reviewer for the question regarding our setting. However, we believe the reviewer has some misunderstanding regarding our settings. Our goal is not simply to transfer the model to unseen subjects; rather, we aim to develop a model that is inherently **subject-agnostic**. Furthermore, the principle of subject-agnostic modeling is **embedded throughout our entire architecture**.
>
> - **Subject-agnostic modeling is much more important for real-world applications**
>
>     Most existing approaches are designed for subject-specific settings, which greatly limits their deployment in real-world scenarios, particularly since most test cases involve new or unseen patients. When applied to a new patient, these models often require more than **12 hours of subject-specific training**, making them impractical for real-world use. Consequently, developing subject-agnostic models that can be directly applied to unseen patients without any additional fine-tuning or retraining is crucial, yet this direction has been largely overlooked by the community. In this work, we aim to develop a **subject-agnostic model** capable of learning subject-agnostic features that can be **directly utilized** for new patients without any fine-tuning or retraining.
>
> - **Our framework is designed to achieve subject-agnostic ability.**
>
>     Our method is explicitly designed to achieve subject-agnostic ability. Our framework consists of three key components—**HCAM, SARA, and HED**—each contributing to this goal from a different perspective.
>
>     HCAM leverages universal cognitive structures to decompose individual cognitive processes, thereby enhancing the model’s understanding of shared cognitive hierarchies across subjects. SARA directly disentangles brain features to extract the core semantic representations that are common to all subjects. HED, in turn, decodes these shared features using a mechanism that is consistent with universal cognitive structures.
>
> Taken together, our setting is directly driven by practical considerations. Building on this foundation, every component of our framework is purposefully designed to support subject-agnostic modeling. Through their coordinated interaction, our method robustly achieves subject-agnostic capability.

---

> ### Author Response · Authors · 2025-11-25
> **Response to Reviewer 9GCm (2/3)**
>
> **W2: about the token extension methods in previous CLIP-MUSED**
>
> We thank the reviewer for mentioning this inspiring work, which will be acknowledged in our revised manuscript. However, our method differs from CLIP-MUSED in several key aspects.
>
> - First, in **CLIP-MUSED**, the token-extension mechanism is explicitly designed to learn **subject-specific features**, which are then used as priors for classification tasks, thereby enhancing the model’s understanding of each individual subject.
>
>     In contrast, in our work, token extension serves a different purpose: it is an integral part of the **SARA** module, whose objective is to **disentangle** the representations into *semantic features* and *subject-specific features*. In this way, our method focuses on extracting **subject-agnostic semantic features** while discarding subject-specific variations, enabling robust subject-agnostic reconstruction.
>
>     From a methodological perspective, unlike CLIP-MUSED—which trains a separate subject prior for each subject—our approach trains a **single, universal disentanglement module**, namely SARA. Once trained, this module can be directly applied to **unseen subjects** to perform disentanglement and extract universal semantic features.
>
>     This also directly addresses Question 4 regarding how subject-specific tokens should be obtained for unseen subjects. In our framework, **no subject-specific tokens need to be pre-trained** for new subjects, because we do not rely on training subject-specific priors. Instead, the disentanglement module itself applies to unseen subjects and produces the required representations without any additional subject-specific training.
>
> - Second, beyond the SARA module, our work also introduces **HCAM and HED**. ****The key idea of **HCAM** is to construct a universal cognitive structure. This design further strengthens the model’s ability to understand information across multiple cognitive dimensions and enhances its cross-subject generalization. In addition, through the **HED** module, we integrate these multi-level features and perform subject-agnostic decoding and reconstruction. **This also differs from CLIP-MUSED,** which primarily relies on aligning representations with a similarity matrix to improve classification performance.
>
> In summary, although SARA shares a superficial resemblance to the token-extension mechanism used in CLIP-MUSED, the two methods differ in both **their goals** and **their underlying approaches**. Furthermore, our framework incorporates additional modules—such as HCAM and HED—to further enhance subject-agnostic ability, which also sets our method apart from CLIP-MUSED.
>
> **W3: about methodological and experimental descriptions in the manuscript**
>
> We sincerely apologize for any confusion caused during the presentation. We will refine our writing accordingly, and this clarification will also be addressed in our response to the questions.
>
> **W4: about the citation form**
>
> Thank you very much for the reminder. We have revised the manuscript accordingly.
>
> **W5: about the reconstruction results**
>
> Thank you for your suggestion. Video reconstruction is inherently challenging, and achieving subject-agnostic reconstruction further increases the difficulty. Therefore, our primary goal is to preserve the core semantic dimensions as faithfully as possible, even if some lower-level details cannot be fully recovered under such constraints.
>
> From a qualitative perspective, the reconstructions exhibit clear semantic correspondence:
>
> - In Figure 1, it captures the concept of *“two people walking along a seaside beach.”*
> - In Figure 2, it captures the concept of *“an airplane flying in a blue sky with clouds.”*
> - In Figure 3, it captures the concept of *“A couple sweetly leaning against each other”*
> - In Figure 4, it captures the concept of *“A person in a swimming pool”*
>
> In terms of overall semantic consistency and completeness, we believe the results demonstrate strong subject-agnostic reconstruction capability. While certain fine-grained, low-level details are indeed difficult to preserve—an expected limitation given the nature of the task—we nonetheless observe meaningful retention of lower-dimensional information, particularly in **color composition** and **global contour structure**.

---

> ### Author Response · Authors · 2025-11-25
> **Response to Reviewer 9GCm (3/3)**
>
> **Q1: about F**
>
> Thank you for pointing this out. Here, **F** indeed refers to the output before the diffusion prior. We recognize that our notation may have caused some ambiguity. We have revised both the equation and the corresponding figure to make the meaning clearer.
>
> **Q2: about the InfoNCE loss**
>
> The objective here is to ensure that the semantic tokens extracted from each subject possess **cross-subject generalization capability**. Although the video dataset contains only two subjects—making this aspect appear less intuitive—the subject-agnostic ability of this module is also learned during the pretraining stage on the image dataset, where multiple subjects are available. This allows the model to acquire robust cross-subject semantic representations before being applied to the video setting.
>
> **Q3: about z and y in Equation 6**
>
> We apologize for not providing sufficient clarification in the original text. Specifically, $y\_{\text{subj}}^{(k)}$ represents the one-hot ground-truth label for the subject, while $z^{(k)}$ denotes the classifier’s logit score corresponding to the probability that the input belongs to subject $k$. We have now added the appropriate annotations in the manuscript to make this definition clear.
>
> **Q4: about inference process**
>
> Thank you for your question, please refer to w2.

---

> > ### Comment · Reviewer_9GCm · 2025-11-26
> >
> > Thank you for your reply.
> >
> > The authors' response has deepened my understanding of their method. The proposed approach does not involve any fine-tuning at all, and it directly applies the model trained on other subjects to a new subject. Even with this clarification, my core concern remains unchanged:
> >
> > 1. In my view, the two designs presented in the paper (Sections 3.1 and 3.3) likewise make no contribution to addressing the adaptation to unseen subjects. Specifically, Section 3.1 designs a hierarchical fMRI representation model inspired by the layered organization of the human visual system, while Section 3.3 introduces captioning, object classification, and segmentation tasks to enable end-to-end training of the entire model. In my view, none of these designs can, in principle, help improve the model’s performance on unseen subjects.
> >
> > 2. If the model directly takes the fMRI data of a new subject as input, how is the difference in the number of voxels across subjects’ ROI handled?

---

> > > ### Author Response · Authors · 2025-11-27
> > > **Response to Reviewer 9GCm**
> > >
> > > Thank you for the follow-up question. We believe the key misunderstanding lies in how our fMRI data are preprocessed and how our architecture uses this representation to achieve subject-invariant decoding.
> > >
> > > **(1) Regarding voxel differences across subjects**
> > >
> > > Our model does *not* directly take raw voxel arrays from each subject. All fMRI volumes are first projected onto the **32k_fs_LR cortical surface**, establishing vertex-wise anatomical correspondence across individuals. We then apply vertex-wise z-normalization and extract the **same set of 8,405 visual-cortex vertices** using the HCPMMP atlas [1]. These vertices exist at fixed indices for all subjects and are finally reshaped into a **1-channel 256×256 map**.
> > >
> > > Therefore, after preprocessing, **every subject’s fMRI data have identical spatial layout, identical vertex count, and consistent topology**, eliminating the voxel-number discrepancy before model training.
> > >
> > > **(2) For Question 1**, we respectfully believe the concern arises from a misunderstanding of what drives subject-agnostic generalization in fMRI-to-vision decoding. The key challenge is that, for the *same* visual stimulus, CLIP consistently produces the **same** semantic representation, whereas different subjects’ fMRI responses vary substantially. Therefore, the central problem in subject-agnostic decoding is to learn a **subject-invariant alignment** that maps heterogeneous fMRI responses to the *universal* CLIP semantic space. Prior cross-subject decoding works (MindEye2 [2], NeuroPictor [3]) have already demonstrated that stronger alignment directly leads to better generalization.
> > >
> > > However, existing approaches are suboptimal because the alignment they learn is **implicit**—the model must infer voxel-to-semantics correspondence without leveraging the well-established topographic organization shared across human visual cortices, nor the hierarchical structure embedded in CLIP. This is where our framework fundamentally differs.
> > >
> > > Because all subjects’ fMRI data are projected into a **unified cortical-surface space**, we are able to incorporate neuroscientific priors into the alignment process. The HCAM module (Section 3.1) explicitly assigns different cortical regions to corresponding CLIP semantic levels, enabling the model to learn **universal hierarchical correspondences** rather than subject-specific mappings. This is precisely the mechanism that improves subject-agnostic performance.
> > >
> > > Furthermore, the HED module (Section 3.3) is designed to **strengthen and stabilize this hierarchical alignment** by introducing explicit tasks—captioning, object classification, and segmentation—that enforce consistency across semantic levels. These tasks are not auxiliary additions; they directly enhance the subject-invariant alignment that ultimately determines the ability to generalize to unseen individuals.
> > >
> > > In short, both HCAM and HED are not generic architectural choices—they are **purposefully constructed to improve alignment**, which is exactly the core prerequisite for generalization on unseen subjects.
> > >
> > > [1] Glasser, Matthew F., et al. "The minimal preprocessing pipelines for the Human Connectome Project." Neuroimage 80 (2013): 105-124.
> > >
> > > [2] Scotti, Paul S., et al. "Mindeye2: Shared-subject models enable fmri-to-image with 1 hour of data." arXiv preprint arXiv:2403.11207 (2024).
> > >
> > > [3] Huo, Jingyang, et al. "Neuropictor: Refining fmri-to-image reconstruction via multi-individual pretraining and multi-level modulation." European Conference on Computer Vision. Cham: Springer Nature Switzerland, 2024.

---

> > > > ### Comment · Reviewer_9GCm · 2025-11-27
> > > >
> > > > Thanks for the author’s further reply.
> > > >
> > > > Regarding voxel differences: By adopting the voxel preprocessing strategy used in fMRI-PTE, one can incorporate anatomy-based priors, allowing the model to achieve a degree of “unseen-subject generalization” already at the input stage. I believe this approach is reasonable.
> > > >
> > > > Regarding subject-invariant alignment:
> > > > Based on the author’s feedback, I understand what they intended to convey: since all subjects in this fMRI dataset were exposed to exactly the same visual stimuli, the authors applied stronger alignment supervision (e.g., contrastive learning between fMRI signals from different brain regions and representations from different layers of the CLIP model, as well as other task-related supervision) to improve representation learning. Ideally, when training on subjects 1 and 2, such an approach might allow the model to learn subject-invariant representations from their fMRI data. However, the model should only be able to extract such invariant representations from the fMRI signals of subjects 1 and 2, because the fMRI patterns of subject 3 are entirely unseen by the model—even when the fMRI-PTE preprocessing strategy is applied. I believe the generalization from subjects 1&2 to subject 3 is highly unlikely based on current methods.

---

> > > > > ### Comment · Reviewer_9GCm · 2025-11-27
> > > > >
> > > > > For the remained issue, I suggest that the authors could validate it through two simple experiments:
> > > > >
> > > > > 1. Using the fMRI-PTE–prealigned voxels from all three subjects, train a simple classification model to predict which subject each fMRI sample comes from. If this classifier can be trained successfully, it would indicate that the preprocessed fMRI data still exhibit substantial inter-subject differences, making them unsuitable for direct generalization in the experimental setup (e.g., 1&2 → 3).
> > > > >
> > > > > 2. Using visualization techniques such as t-SNE or UMAP, one could visualize the spatial distribution of the fMRI representations from the three subjects. This would help assess whether subject-invariant representations can truly be extracted by the model.

---

> > > > > > ### Author Response · Authors · 2025-11-27
> > > > > > **Response to Reviewer 9GCm**
> > > > > >
> > > > > > Thank you for the helpful suggestions. We would like to clarify that anatomy-based preprocessing only provides the basis for unseen-subject generalization—i.e., ensuring that all subjects share the same spatial layout. At this stage, the data are **not** expected to be subject-invariant in the feature level, because no learning has occurred; preprocessing simply applies offline surface-based transformations. As shown in Fig. 9 in Appendix C.5, a t-SNE visualization of the preprocessed voxels indeed produces clear subject-specific clusters, which is fully expected. Moreover, we also conducted a classification experiment following your suggestion. Specifically, we flattened the preprocessed data and trained a simple 3-class MLP classifier. The results are as follows:
> > > > > >
> > > > > > ```
> > > > > > Epoch [1/10]  Train Loss: 0.3451  Train Acc: 0.9020  Test Acc: 0.9456
> > > > > > Epoch [2/10]  Train Loss: 0.0031  Train Acc: 0.9998  Test Acc: 0.9553
> > > > > > Epoch [3/10]  Train Loss: 0.0010  Train Acc: 1.0000  Test Acc: 0.9614
> > > > > > Epoch [4/10]  Train Loss: 0.0006  Train Acc: 1.0000  Test Acc: 0.9642
> > > > > > Epoch [5/10]  Train Loss: 0.0004  Train Acc: 1.0000  Test Acc: 0.9656
> > > > > > Epoch [6/10]  Train Loss: 0.0004  Train Acc: 1.0000  Test Acc: 0.9686
> > > > > > Epoch [7/10]  Train Loss: 0.0003  Train Acc: 1.0000  Test Acc: 0.9611
> > > > > > Epoch [8/10]  Train Loss: 0.0002  Train Acc: 1.0000  Test Acc: 0.9647
> > > > > > Epoch [9/10]  Train Loss: 0.0001  Train Acc: 1.0000  Test Acc: 0.9689
> > > > > > Epoch [10/10] Train Loss: 0.0001  Train Acc: 1.0000  Test Acc: 0.9733
> > > > > > Final Test Accuracy: 0.9733
> > > > > > ```
> > > > > >
> > > > > > These results are consistent with the t-SNE visualization.
> > > > > >
> > > > > > The **true subject-invariant representation** emerges in the **high-dimensional feature space only after model training**. With the HCAM and HED modules enforcing hierarchical semantic alignment, the learned invariant features show a completely different pattern. As shown in Fig. 9 in Appendix C.5, the t-SNE visualization of the semantic token space ($\mathbf{T}_{\text{sem}} \in \mathbb{R}^{B \times S \times L \times C}$) reveals that samples from all subjects are **well mixed**, indicating that subject-specific variations have been effectively reduced.
> > > > > > This contrast between the visualization of the **preprocessed but non-invariant data** and the visualization of the **learned, subject-invariant semantic tokens** directly demonstrates the effectiveness of our method.

---

> ### Comment · Reviewer_9GCm · 2025-11-28
>
> Thank you for the response.
>
> The additional visualization results indeed show that, even though the model is directly applied to unseen subjects without any adaptation, it still appears capable of extracting subject-invariant representations. Even though I still have some concerns about the novelty and interpretability of the method, considering the validity of the experimental results and the efforts the authors have made during the discussion period, I feel it is necessary to slightly increase my rating.

---

> > ### Author Response · Authors · 2025-11-28
> > **Response to Reviewer 9GCm**
> >
> > Thank you for raising your score; we are glad to hear that most of your concerns have been addressed. We also appreciate your engagement with our rebuttal and your prompt response. We will do our best to address any remaining concerns during the discussion and are happy to provide further clarification if needed.

---

> > > ### Comment · Reviewer_9GCm · 2025-11-28
> > >
> > > Thanks for your responds, I now have a good understanding of the methods in this manuscript. I would like to change my rating to 6, but I found that the system no longer seems to allow rating revisions, which might be an OpenReview bug. I will report my final rating to AC in subsequent discussions.

---

### Author Response · Authors · 2025-11-29
**General Response**

We sincerely thank all reviewers for their valuable feedback. Regarding the **motivation** of our work, reviewers consistently recognized its strength, noting that it `focuses on an important issue` (**9GCm**), is `clearly explained and detailed` (**HxmM**), and that `the motivation of the article is very good` (**y215**).
Our **overall framework design** was also well received and acknowledged by multiple reviewers. They appreciated that our method `not only designs multiple different modules to extract features at different levels for alignment, but also adopts multiple tasks to conduct a more comprehensive alignment`.(**y215**). In particular, reviewers highlighted that `the proposed SARA module and fine-grained hierarchical alignment mechanism achieve good performance improvements in the cross-subject setting` (**ebCw**). Furthermore, reviewers appreciated the **clarity of our writing** (**HxmM**, **y215**) and praised both the **strong performance of our experiments** (**ebCw**) and the **thoroughness of our experimental evaluation** (**ebCw**, **y215**).

The reviewers’ main concerns center around clarifications of our methodology, questions regarding implementation details, and requests for additional experiments.

We have carefully addressed all reviewer comments and revised the manuscript accordingly. After discussing with reviewer **9GCm**, the reviewer decided to **raise the score**.

A revised version has been uploaded with all changes highlighted in blue.

---

### Meta-Review · Area_Chair_Sdav · 2026-01-06

**Summary:**

This manuscript contributes an fMRI-to-video reconstruction, with a focus on inter-subject features.
The reviewers overall appreciated the good motivation and progress on an interesting topic. They did bring up limited novelty, given that a similar technique had already been applied to fMRI-to-image reconstruction.
Likewise, they appreciated the comprehensives ablations, but found the resulting evidence muddy.
Finally, they brought up the limited number of subjects (3), which undermines a clear demonstration.

**Reviewer Concerns:**

There was a discussion cross-subject transfer versus subject-invariant features. The authors stated that the goal is subject-agnostic learning, and not transferring to new subjects (though the AC doesn't see how these goals could differ). New visualization convinced the reviewer that the model was indeed learning subject-invariant features.

**Reviewer Scores:**

One reviewer did mention raising scores from 4 to 6

---

### Decision · Program_Chairs · 2026-01-26

Accept (Poster)